# A graph-based network for predicting chemical reaction pathways in solid-state materials synthesis

Matthew J. McDermott[1,2], Shyam S. Dwaraknath [1] & Kristin A. Persson [2,3✉]

Accelerated inorganic synthesis remains a significant challenge in the search for novel, functional materials. Many of the principles which enable "synthesis by design" in synthetic organic chemistry do not exist in solid-state chemistry, despite the availability of extensive computed/experimental thermochemistry data. In this work, we present a chemical reaction network model for solid-state synthesis constructed from available thermochemistry data and devise a computationally tractable approach for suggesting likely reaction pathways via the application of pathfinding algorithms and linear combination of lowest-cost paths in the network. We demonstrate initial success of the network in predicting complex reaction pathways comparable to those reported in the literature for $YMnO_3$, $Y_2Mn_2O_7$, $Fe_2SiS_4$, and $YBa_2Cu_3O_{6.5}$. The reaction network presents opportunities for enabling reaction pathway prediction, rapid iteration between experimental/theoretical results, and ultimately, control of the synthesis of solid-state materials.

[1] Materials Sciences Division, Lawrence Berkeley National Laboratory, Berkeley, CA, USA. [2] Department of Materials Science and Engineering, University of California, Berkeley, CA, USA. [3] Molecular Foundry, Lawrence Berkeley National Laboratory, Berkeley, CA, USA. ✉email: kapersson@lbl.gov

D ating back to 18th century mineralogy[1], solid-state inorganic chemistry is a cornerstone in the design of novel, functional materials and continues to be driven by pressing technological demands. Consequently, the development of new techniques that accelerate materials synthesis/processing is vital for achieving multifunctional materials with complex properties. Solid materials with target functionality are often thermodynamically metastable, which can limit their accessibility via conventional solid-state synthesis routes, such as the classic "shake and bake" ceramic methods that typically require high temperatures to overcome diffusion barriers and often proceed to global thermodynamic equilibrium[2]. Indeed, solid-state chemistry itself has even been dubbed a black box that is best probed via systematic and extensive iteration, requiring significant experimental expertise akin to apprenticed artistry[3]. The optimization of synthesis procedures for new materials is hence both highly time-consuming and resource-consuming, demanding human-guided iteration over many combinations of precursors, processing steps, and environmental conditions.

A more efficient approach to synthesizing novel inorganic materials is "synthesis by design", in which a set of guiding principles is used to quickly devise a synthesis method towards a target material, much like the paradigm central to synthetic organic chemistry[4,5]. Recent work, fueled by developments in solid-state in situ characterization techniques[6,7], has advanced this direction by exploring reaction pathways in select case systems that provide insight into mechanistic relationships explaining how synthesis conditions (e.g., precursor selection, reaction environment) alter the reaction pathway and lead to selective formation of different target products. For example, Neilson and coworkers demonstrated the use of unconventional solid-state metathesis reactions to kinetically control the reaction pathway towards metastable polymorphs of $CuSe_2$[8] and $YMnO_3$[9,10]. Jiang et al. explored the use of iron silicide reactants to bypass kinetic limitations and achieve a low-temperature synthesis of $Fe_2SiS_4$[11]. Miura et al. demonstrated the synthesis of $MgCr_2S_4$ thiospinel via a metathesis route using novel precursors, which was shown to be thermodynamically favorable through computational phase diagram construction[12]. Bianchini et al. showed that the first phase formed in the synthesis of P2-type $Na_{0.67}MO_2$ (M=Co, Mn) can be predicted by minimizing compositionally unconstrained reaction energies and that the initial phase formed may drastically alter both the kinetics of subsequent reactions, as well as final phase selectivity[13]. Each of these studies elucidates an important concept: chemical reaction pathways follow a complex thermodynamic free energy landscape that can be carefully manipulated and navigated via the thoughtful selection of precursors, processing, and environmental conditions.

Explicit modeling, as well as reaction network models derived from the atomistic potential energy surface (PES), have been successful in predicting chemical reaction pathways in molecular systems[14,15] but are much less developed for solid-state periodic systems, where monitoring each atom's coordinates and interactions over the large time and spatial scales necessary rapidly becomes intractable. Despite these limitations, modeling of bounded solid-state reaction mechanisms at the atomistic level has been achieved in particular with molecular dynamics (MD)[16] and kinetic Monte Carlo (KMC)-based[17] approaches. Reactive force fields, such as ReaxFF[18] further permit the breaking of chemical bonds and can be used to study specific chemical reaction mechanisms and kinetic parameters[19]. KMC-based methods also explore parts of the PES, given reaction rate constants that can be approximated with quantum mechanical calculations. However, such methods are ultimately confined to an a priori selection of the relevant domains of the high dimensional solid-state PES. Recent work also suggests that the computational prediction of reaction pathways in ceramic powder-based synthesis does not always require atomistic methods; significant predictive power can be derived from local thermodynamic equilibrium calculations of pairwise solid-solid interfaces[20].

In this work, we describe a chemical reaction network framework for predicting and suggesting solid-state inorganic reaction pathways, which when combined with experimental efforts, aims to realize inorganic synthesis by design. We propose to leverage recent advances in data-driven methods that have resulted in computational/experimental thermochemistry databases[21–24] covering hundreds of thousands of materials and millions of associated reaction energies[25]. We employ a reaction network model that blends typical thermodynamic phase diagrams with the connectivity and kinetic heuristics derived from transition state theory. The network model serves as a convenient data structure for exploring the underlying free energy surface of thermodynamic phase space in solid-state chemistry via the power and efficiency of existing computational infrastructure for large graph networks. We outline the methodology used to create the chemical reaction network from thermochemistry databases and demonstrate its capacity for solid-state reaction pathway prediction by applying it to several reported experimental syntheses, as well as to recommend chemical routes to a novel battery cathode material that has not been previously synthesized.

## Results

The solid-state reaction network is a model for thermodynamic phase space, which is represented by an energy landscape governed by a generalized thermodynamic potential or free energy, $\Phi$. The global minimum in this potential, which depends on the boundary conditions of a particular system, is the thermodynamic equilibrium state of the system. Figure 1 depicts three models of chemical reactions in thermodynamic phase space, ordered by increasing the level of abstraction. The free energy convex hull construction of Fig. 1a is a purely thermodynamic model of a chemical reaction between two reactant phases, $R_1$ and $R_2$[26]. The convex hull yields the set of products (and thus chemical reactions) that result in the largest decrease in free energy for a given mole ratio of the two reactants. Figure 1b abstracts the thermodynamic model further by incorporating the concept of activation energy, $E_a$, as defined by transition state theory[27]. This enables the inclusion of simple kinetic behavior of reactions, where the height of the activation energy barrier correlates with the rate of reaction.

Abstracting even further, we can consider these reaction coordinate diagrams as weighted directed graphs, like that shown in the upper portion of Fig. 1b. In these graphs, the cost/weight of a chemical reaction edge represents an a priori unknown function of synthesis parameters such as the thermodynamic driving force, activation energy, etc. Figure 1c shows the interlinking of many such graph representations within a set of phases, where each node represents a particular combination of phases (e.g., $R_1 + R_2$) and the edges represent chemical reactions with designated costs. This weighted directed graph, or chemical reaction network, is a densely connected model of thermodynamic phase space where thermodynamic/kinetic features can be combined and transformed into a unique cost representation for each reaction pathway.

In the following sections, the chemical reaction network method is applied to solid-state synthesis procedures discussed in the literature for $YMnO_3$, $Y_2Mn_2O_7$, $Fe_2SiS_4$, and $YBa_2Cu_3O_{6.5}$. The reaction networks generated for each experimental system are illustrated in Fig. 2 and constructed primarily using thermochemistry data acquired from the Materials Project (MP)[21]. Via the application of pathfinding algorithms and several

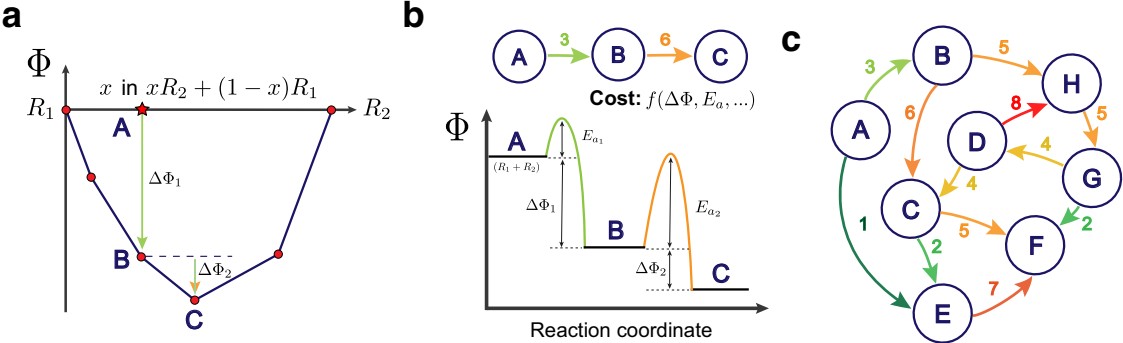

**Fig. 1 Thermodynamic models of chemical reactions in increasing degree of abstraction.** The label, *A*, indicates a combination of arbitrary amounts of phases $R_1$ and $R_2$. Similarly, the other labels (*B–F*) are combinations of arbitrary amounts of other phases that result from the chemical reactions. **a** The convex hull construction for consecutive reactions between two reactants, $R_1$ and $R_2$. The points drawn indicate sets of chemical reaction products at different stoichiometric mixtures of the reactants, and the lines trace the convex hull indicating the thermodynamic equilibrium (minimum free energy) for all ratios of mixing, *x*. **b** A traditional reaction coordinate diagram used to represent both the free energies of reaction, $\Delta\Phi$, and activation energies, $E_a$. This is generalized by a weighted, directed graph connecting the three states (top). The cost/weight of the directed edges, shown as the colored number adjacent to each edge, is some function of the free energy change, activation energy, and other reaction features. **c** A chemical reaction network, which links together many such possible reaction pathways within a given set of phases.

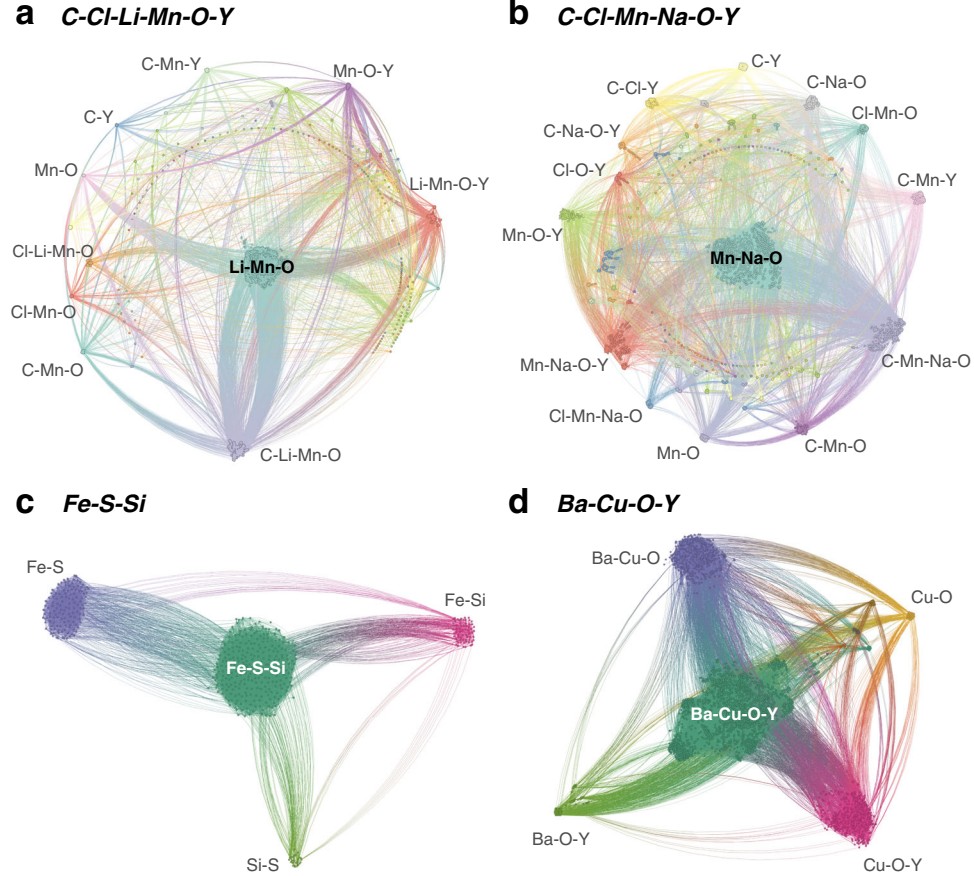

**Fig. 2 Visualizations of several computed reaction networks.** Chemical subsystems are illustrated by color and prominent subsystems are annotated with labels. Shown are reaction networks for the chemical systems: **a** C–Cl–Li–Mn–O–Y with 5855 nodes and 121,176 edges, **b** C–Cl–Mn–Na–O–Y with 4425 nodes and 46,427 edges, **c** Fe–S–Si with 509 nodes and 11,912 edges, and **d** Ba–Cu–O–Y with 2973 nodes and 33,957 edges.

post-processing steps (see "Methods"), we predict the top-ranked candidate reaction pathways to each of the targets and compare them with previous experimental results. Finally, we apply the network construction to predict synthesis routes towards a novel battery cathode material that has yet to be synthesized, $MgMo_3(PO_4)_3O$, further demonstrating the versatility of the method in advancing inorganic synthesis by design.

**Synthesis of YMnO₃ using Li-based assisted metathesis.** First, we consider the synthesis of yttrium manganese oxide (YMnO₃) through the solid-state Li-based assisted metathesis reaction first reported by Todd and Neilson[9]. This synthesis route has the advantage of yielding YMnO₃ at temperatures significantly lower than the reaction between binary oxides (500 °C vs. 850 °C), enabling kinetic control and polymorph selectivity.

The overall reaction,

$$Mn_2O_3 + 2\ YCl_3 + 3\ Li_2CO_3 \rightarrow 2\ YMnO_3 + 6\ LiCl + 3\ CO_2 \tag{1}$$

proceeds through several steps with distinct intermediate compounds, as determined experimentally through in situ temperature-dependent x-ray diffraction performed at a synchrotron beamline[10].

Figure 2a shows the chemical reaction network generated for the C–Cl–Li–Mn–O–Y chemical system. The phase diagram for this system includes 853 entries; of these, 53 are predicted by density functional theory (DFT) to be stable at low temperatures. By incorporating vibrational entropic effects through a previously derived machine-learning methodology (see "Methods" section), we find the number of stable species to reduce to 41 at a temperature of 900 K. We include all of these stable entries, as well as metastable entries, not including polymorphs, up to a filter of +30 meV/atom above the hull. This cutoff is motivated by DFT calculations and statistics on experimentally available phases[28] showing that—while highly metastable compounds are by no means inaccessible—the distribution of energies above hull for all experimentally synthesized compounds peaks significantly below 30 meV/atom. This process results in a total of 76 phases considered (Supplementary Data 1) which yields a reaction network of 5855 nodes and 121,176 edges. Reaction edge costs were calculated using the softplus function applied to reaction free energies normalized by the number of reactant atoms. The 60 shortest paths (20 to each of the products YMnO₃, LiCl, and CO₂) were identified, and crossover reactions were generated considering oxygen as an open-element with a chemical potential of $\mu_O = \mu_O^{exp}(900\ K,\ 1\ atm)$[22]. The final reaction pathways were predicted via this candidate set of reactions by solving for all possible (mass-balanced) linear combinations of reactions up to a maximum size of five reaction steps. This resulted in 38 pathways, which are listed in full in Supplementary Data 1. This list further reduces to 20 pathways after removing pathways with interdependent reaction steps (see "Methods" section).

Of the 20 remaining total reaction pathways, 11 paths involve reaction steps that produce one or more of the following hypothetical intermediate compounds that, to our best knowledge, have never been experimentally synthesized: $Li_3MnO_3$, $Li_2MnCO_4$, and $Li_2MnCO_5$. Furthermore, each of these phases is predicted to be metastable on the MP database, with energies above the hull of 17, 98, and 35 meV/atom, respectively. Interestingly, the lowest cost predicted pathway (before filtering) actually proceeds through the hypothetical $Li_3MnO_3$ as follows:

$$1.5\ Li_2CO_3 + 0.5\ Mn_2O_3 \rightarrow Li_3MnO_3 + 1.5\ CO_2$$
$$(\Delta G_{rxn} = 0.079\ eV/atom) \tag{2}$$

$$YCl_3 + Li_3MnO_3 \rightarrow YMnO_3 + 3LiCl$$
$$(\Delta G_{rxn} = -0.223\ eV/atom) \tag{3}$$

This pathway through $Li_3MnO_3$, as well as similar ones that rely on the formation of hypothetical compounds, may provide useful insights in understanding the lack of accessibility of certain phases, but are deemed less likely to suggest experimentally viable reaction pathways. The five lowest-cost pathways, after removing the pathways which pass through hypothetical intermediates, are summarized in Table 1.

Paths 2 and 3 are identical to the experimentally observed low-temperature reaction pathway reported by Todd and Neilson[10], involving the formation and reaction of ternary intermediates $YOCl + LiMnO_2 \rightarrow YMnO_3 + LiCl$, with the only difference being that Path 3 first includes the decomposition of $Li_2CO_3$ to $Li_2O$ before subsequent reaction with $YCl_3$. Note that Path 2 includes a reaction with three products (even though the network was restricted to a maximum combination size of $n = 2$) due to the calculation of crossover reactions, which uses the compositional phase diagram to find the most thermodynamically favorable set of products (of any size). Paths 1, 4, and 5 also encompass a plausible reaction pathway and differ primarily in the manner in which $Y_2O_3$ is formed and subsequently reacted with $Mn_2O_3$ to produce the final target $YMnO_3$. Todd et al. report that the oxidation of $YCl_3 \rightarrow YOCl \rightarrow Y_2O_3$ and reaction of

**Table 1 Top five predicted reaction pathways to YMnO₃.**

| Path | Total cost | Reactions | $\Delta G_{rxn}$ (eV/atom) | Cost |
|---|---|---|---|---|
| 1 | 0.246 | $0.6667\ Li_2O + 0.3333\ YCl_3 \rightarrow LiCl + 0.3333\ LiYO_2$ | −0.260 | 0.210 |
| | | $0.5\ YCl_3 + 1.5\ LiYO_2 \rightarrow Y_2O_3 + 1.5\ LiCl$ | −0.164 | 0.229 |
| | | $0.5\ Y_2O_3 + 0.5\ Mn_2O_3 \rightarrow YMnO_3$ | −0.048 | 0.254 |
| | | $Li_2CO_3 \rightarrow Li_2O + CO_2$ | 0.145 | 0.300 |
| 2 | 0.249 | $Li_2CO_3 + YCl_3 \rightarrow YOCl + 2\ LiCl + CO_2$ | −0.110 | 0.240 |
| | | $YOCl + LiMnO_2 \rightarrow LiCl + YMnO_3$ | −0.069 | 0.249 |
| | | $Li_2CO_3 + Mn_2O_3 \rightarrow CO_2 + 2\ LiMnO_2$ | 0.007 | 0.266 |
| 3 | 0.254 | $Li_2O + YCl_3 \rightarrow YOCl + 2\ LiCl$ | −0.281 | 0.206 |
| | | $YOCl + LiMnO_2 \rightarrow LiCl + YMnO_3$ | −0.069 | 0.249 |
| | | $Li_2CO_3 + Mn_2O_3 \rightarrow CO_2 + 2\ LiMnO_2$ | 0.007 | 0.266 |
| | | $Li_2CO_3 \rightarrow Li_2O + CO_2$ | 0.145 | 0.300 |
| 4 | 0.260 | $4\ Li_2O + 3\ YCl_3 \rightarrow Y_3O_4Cl + 8\ LiCl$ | −0.297 | 0.203 |
| | | $0.6667\ Li_2O + 0.3333\ YCl_3 \rightarrow LiCl + 0.3333\ LiYO_2$ | −0.260 | 0.210 |
| | | $0.5\ Y_3O_4Cl + 0.5\ LiYO_2 \rightarrow Y_2O_3 + 0.5\ LiCl$ | −0.057 | 0.252 |
| | | $0.5\ Y_2O_3 + 0.5\ Mn_2O_3 \rightarrow YMnO_3$ | −0.048 | 0.254 |
| | | $Li_2CO_3 \rightarrow Li_2O + CO_2$ | 0.145 | 0.300 |
| 5 | 0.267 | $3\ Li_2O + 2\ YCl_3 \rightarrow Y_2O_3 + 6\ LiCl$ | −0.306 | 0.202 |
| | | $0.6667\ Li_2O + 0.3333\ YCl_3 \rightarrow LiCl + 0.3333\ LiYO_2$ | −0.260 | 0.210 |
| | | $0.5\ YCl_3 + 1.5\ LiYO_2 \rightarrow Y_2O_3 + 1.5\ LiCl$ | −0.164 | 0.229 |
| | | $0.5\ Y_2O_3 + 0.5\ Mn_2O_3 \rightarrow YMnO_3$ | −0.048 | 0.254 |
| | | $Li_2CO_3 \rightarrow Li_2O + CO_2$ | 0.145 | 0.300 |

These are the lowest total cost reaction pathways after filtering for interdependent reaction steps and previously unsynthesized compounds. Paths 2 and 3 match the experimentally reported pathway most closely, however, the oxidation of $YCl_3 \rightarrow YOCl \rightarrow Y_2O_3$ and subsequent reaction with $Mn_2O_3$ in Paths 1, 4, and 5 were experimentally suggested as a possible route, albeit much slower and only kinetically accessible at higher temperatures.

**Table 2 Selection of top predicted reaction pathways to $Y_2Mn_2O_7$.**

| Path | Total cost | Reactions | $\Delta G_{rxn}$ (eV/atom) | Cost |
|---|---|---|---|---|
| 1 | 0.238 | $Na_2CO_3 + YCl_3 \rightarrow CO_2 + YOCl + 2\,NaCl$ | −0.178 | 0.226 |
| | | $0.5\,O_2 + 2\,YOCl + 2\,NaMnO_2 \rightarrow Y_2Mn_2O_7 + 2\,NaCl$ | −0.164 | 0.229 |
| | | $Mn_2O_3 + Na_2CO_3 \rightarrow CO_2 + 2\,NaMnO_2$ | 0.024 | 0.271 |
| 2 | 0.246 | $2.125\,O_2 + 0.1667\,YMn_{12} + 0.9167\,Y_2O_3 \rightarrow Y_2Mn_2O_7$ | −0.926 | 0.113 |
| | | $0.5\,O_2 + Na_4Mn_2O_5 + 4\,YOCl \rightarrow Y_2Mn_2O_7 + Y_2O_3 + 4\,NaCl$ | −0.222 | 0.217 |
| | | $Na_2CO_3 + YCl_3 \rightarrow CO_2 + YOCl + 2\,NaCl$ | −0.178 | 0.226 |
| | | $0.5\,Mn_2O_3 + Na_2CO_3 \rightarrow CO_2 + 0.5\,Na_4Mn_2O_5$ | 0.120 | 0.294 |
| | | $1.308\,Mn_2O_3 + 1.026\,Y_2O_3 \rightarrow Y_2Mn_2O_7 + 0.05128\,YMn_{12}$ | 0.144 | 0.300 |
| 3 | 0.246 | $0.5\,O_2 + Na_4Mn_2O_5 + 4\,YOCl \rightarrow Y_2Mn_2O_7 + Y_2O_3 + 4\,NaCl$ | −0.222 | 0.217 |
| | | $Na_2CO_3 + YCl_3 \rightarrow CO_2 + YOCl + 2\,NaCl$ | −0.178 | 0.226 |
| | | $2\,MnO_2 + Y_2O_3 \rightarrow Y_2Mn_2O_7$ | −0.070 | 0.249 |
| | | $0.25\,O_2 + 0.5\,Mn_2O_3 \rightarrow MnO_2$ | −0.058 | 0.252 |
| | | $0.5\,Mn_2O_3 + Na_2CO_3 \rightarrow CO_2 + 0.5\,Na_4Mn_2O_5$ | 0.120 | 0.294 |
| 4 | 0.248 | $4\,Na_2CO_3 + YCl_3 \rightarrow Na_5Y(CO_3)_4 + 3\,NaCl$ | −0.193 | 0.223 |
| | | $0.5\,O_2 + 2\,YOCl + 2\,NaMnO_2 \rightarrow Y_2Mn_2O_7 + 2\,NaCl$ | −0.164 | 0.229 |
| | | $0.25\,Na_5Y(CO_3)_4 + 0.75\,YCl_3 \rightarrow CO_2 + YOCl + 1.25\,NaCl$ | −0.050 | 0.254 |
| | | $Mn_2O_3 + Na_2CO_3 \rightarrow CO_2 + 2\,NaMnO_2$ | 0.024 | 0.271 |
| 11 | 0.259 | $Na_2O + YCl_3 \rightarrow YOCl + 2\,NaCl$ | −0.535 | 0.164 |
| | | $2\,YOCl + Na_2O \rightarrow Y_2O_3 + 2\,NaCl$ | −0.339 | 0.196 |
| | | $2\,MnO_2 + Y_2O_3 \rightarrow Y_2Mn_2O_7$ | −0.070 | 0.249 |
| | | $0.25\,O_2 + 0.5\,Mn_2O_3 \rightarrow MnO_2$ | −0.058 | 0.252 |
| | | $Na_2CO_3 \rightarrow CO_2 + Na_2O$ | 0.328 | 0.351 |

The complex defect cascade observed in the experiment is not directly present in the predicted pathways, although the reaction of $2MnO_2 + Y_2O_3 \rightarrow Y_2Mn_2O_7$ present in Paths 3 and 11 represents the net effect of the defect cascade reaction, and Path 11 is overall the closest to the experimentally observed pathway.

binary oxides $Y_2O_3 + Mn_2O_3 \rightarrow 2\,YMnO_3$ is also a plausible reaction pathway that may simultaneously occur, although ex situ control reactions showed that it is much slower and only feasible at higher temperatures. While $Li_2O$ (Paths 1, 3–5) and $LiYO_2$ (Path 1, 4, 5) were not directly observed in the diffraction results, all of the other intermediate phases suggested by the top predicted paths ($YOCl$, $Y_3O_4Cl$, $Y_2O_3$, $LiMnO_2$) were indeed observed in experimental data. Hence the reaction network predictions here seem to capture the form of both experimentally observed reaction pathways, suggesting that the model performs quite well with this particular system.

**Synthesis of $Y_2Mn_2O_7$ using Na-based assisted metathesis**. According to the original report by Todd and Neilson, substituting Na for Li in the aforementioned assisted metathesis synthesis of $YMnO_3$ changes the main product to the pyrochlore $Y_2Mn_2O_7$[9], represented by the net reaction:

$$Mn_2O_3 + 2\,YCl_3 + 3\,Na_2CO_3 + 0.5\,O_2 \rightarrow Y_2Mn_2O_7 + 6\,NaCl + 3\,CO_2 \tag{4}$$

It was shown via in situ x-ray diffraction that, similar to the Li-based reaction, the Na-based reaction pathway similarly depends on the formation of an intermediate alkali manganese oxide phase—in this case, $Na_xMnO_2$[29]. Through a cascade of defect reactions, $Na_xMnO_2$ reacts with $Y_2O_3$, as formed through the previously described oxidation of $YCl_3 \rightarrow YOCl \rightarrow Y_2O_3$.

The reaction network for the C–Cl–Mn–Na–O–Y chemical system was constructed with the same parameters as the Li-based network: a temperature of $T = 900$ K and a 30 meV/atom filter, resulting in 66 entries (Supplementary Data 2) mapped to 4425 nodes and 46,427 edges. Figure 2b illustrates the network for this system, which is similar in shape to the Li-based system, but with a higher number of prominent chemical subsystems. Pathfinding was also performed with identical parameters as the Li system, including $k = 20$ paths to each target, a maximum reaction combination size of 5, and an open oxygen chemical potential of $\mu_O = \mu_O^{exp}(900\,K, 1\,atm)$[22]. The pathfinding process resulted in 44

potential reaction pathways (Supplementary Data 2), 35 of which do not contain interdependent reaction steps, and 24 of which further do not contain any reaction steps with the hypothetical phases $Na_3MnO_3$, $Mn_8Cl_3O_{10}$, and $Na_2MnO_3$. A selection of the top predicted pathways are shown in Table 2.

The top predicted pathway (Path 1) is directly analogous to the observed kinetically favorable pathway in the Li-based synthesis, although this was not observed experimentally for the Na-based synthesis. Instead, Paths 3 and 11, which contain the final reaction step $2MnO_2 + Y_2O_3 \rightarrow Y_2Mn_2O_7$, more closely resemble the experimentally observed pathway, which involves a cascade of defect reactions where $MnO_2$ sub-units (anchored within $Na_xMnO_2$) react with $Y_2O_3$, resulting in a steady increase of the effective Na concentration, $x$, as $Y_2Mn_2O_7$ forms.

Interestingly, $Na_4Mn_2O_5$, $Na_5Y(CO_3)_4$, and $YMn_{12}$ appear as intermediate phases in several of the top predicted pathways but were not observed experimentally. While each of these phases is experimentally synthesizable and predicted by DFT as stable, their synthesis in the literature is reportedly difficult and involves long heating times[30], hydrothermal methods[31], or the use of a levitation furnace[32], respectively. Hence these phases may be kinetically inaccessible using ceramic methods, including the assisted metathesis synthesis method discussed here.

**Synthesis of $Fe_2SiS_4$ using iron silicide precursors**. Jiang et al. previously showed that $Fe_2SiS_4$ can be synthesized at significantly lower temperatures by avoiding the kinetically limiting steps encountered when using elemental Fe and Si precursors[11]. They showed that pre-reacting iron with silicon to create iron silicide precursors greatly expedited the formation of $Fe_2SiS_4$ at temperatures as low as 550 °C, via the net reaction:

$$Fe_5Si_3 + Fe_3Si + 16\,S \rightarrow 4\,Fe_2SiS_4 \tag{5}$$

The reaction network for the Fe–S–Si chemical system was constructed at a temperature of $T = 900$ K with much less stringent energy above hull filter of 0.5 eV/atom. This higher energy cutoff was chosen due to the relatively small size of the

**Table 3 Selection of top predicted reaction pathways to $Fe_2SiS_4$.**

| Path | Total cost | Reactions | $\Delta G_{rxn}$ (eV/atom) | Cost |
|---|---|---|---|---|
| 1 | 0.177 | $0.6667\ FeSi_2 + 3.333\ S \rightarrow SiS_2 + 0.3333\ Fe_2SiS_4$ | −0.787 | 0.129 |
| | | $5\ S + Fe_3Si \rightarrow Fe_2SiS_4 + FeS$ | −0.586 | 0.156 |
| | | $0.4286\ Fe_5Si_3 + 4\ S \rightarrow Fe_2SiS_4 + 0.1429\ FeSi_2$ | −0.543 | 0.162 |
| | | $SiS_2 + 2FeS \rightarrow Fe_2SiS_4$ | 0.270 | 0.335 |
| 2 | 0.177 | $0.6667\ FeSi_2 + 3.333\ S \rightarrow SiS_2 + 0.3333\ Fe_2SiS_4$ | −0.787 | 0.129 |
| | | $5\ S + Fe_3Si \rightarrow Fe_2SiS_4 + FeS$ | −0.586 | 0.156 |
| | | $S + 0.4\ Fe_3Si \rightarrow FeS + 0.2\ FeSi_2$ | −0.545 | 0.162 |
| | | $0.4286\ Fe_5Si_3 + 4\ S \rightarrow Fe_2SiS_4 + 0.1429\ FeSi_2$ | −0.543 | 0.162 |
| | | $SiS_2 + 2\ FeS \rightarrow Fe_2SiS_4$ | 0.270 | 0.335 |
| 3 | 0.179 | $0.6667\ FeSi_2 + 3.333\ S \rightarrow SiS_2 + 0.3333\ Fe_2SiS_4$ | −0.787 | 0.129 |
| | | $2\ FeSi + 6\ S \rightarrow SiS_2 + Fe_2SiS_4$ | −0.632 | 0.149 |
| | | $5\ S + Fe_3Si \rightarrow Fe_2SiS_4 + FeS$ | −0.586 | 0.156 |
| | | $0.4286\ Fe_5Si_3 + 4\ S \rightarrow Fe_2SiS_4 + 0.1429\ FeSi_2$ | −0.543 | 0.162 |
| | | $S + 0.5\ Fe_3Si \rightarrow FeS + 0.5\ FeSi$ | −0.509 | 0.167 |
| | | $SiS_2 + 2\ FeS \rightarrow Fe_2SiS_4$ | 0.270 | 0.335 |
| 4 | 0.180 | $0.6667\ FeSi_2 + 3.333\ S \rightarrow SiS_2 + 0.3333\ Fe_2SiS_4$ | −0.787 | 0.129 |
| | | $2\ FeSi + 6\ S \rightarrow SiS_2 + Fe_2SiS_4$ | −0.632 | 0.149 |
| | | $5\ S + Fe_3Si \rightarrow Fe_2SiS_4 + FeS$ | −0.586 | 0.156 |
| | | $0.4286\ Fe_5Si_3 + 4\ S \rightarrow Fe_2SiS_4 + 0.1429\ FeSi_2$ | −0.543 | 0.162 |
| | | $0.5\ Fe_5Si_3 + 4\ S \rightarrow Fe_2SiS_4 + 0.5\ FeSi$ | −0.521 | 0.166 |
| | | $SiS_2 + 2\ FeS \rightarrow Fe_2SiS_4$ | 0.270 | 0.335 |
| 14 | 0.189 | $2\ S + 0.5\ Fe_3Si \rightarrow FeS_2 + 0.5\ FeSi$ | −0.635 | 0.149 |
| | | $2\ FeSi + 6\ S \rightarrow SiS_2 + Fe_2SiS_4$ | −0.632 | 0.149 |
| | | $5\ S + Fe_3Si \rightarrow Fe_2SiS_4 + FeS$ | −0.586 | 0.156 |
| | | $2\ Fe_5Si_3 + 20\ S \rightarrow Si + 5\ Fe_2SiS_4$ | −0.541 | 0.163 |
| | | $Si + 2\ FeS_2 \rightarrow Fe_2SiS_4$ | 0.055 | 0.278 |
| | | $SiS_2 + 2\ FeS \rightarrow Fe_2SiS_4$ | 0.270 | 0.335 |

Each pathway successfully includes the experimentally observed reaction between intermediates $SiS_2$ and FeS. The observed FeSi intermediate only appears together with the other two intermediates in 4 of the 49 predicted paths. Path 14 is the lowest cost path which captures all experimentally observed intermediate phases.

chemical system (3 elements), which allowed for consideration of a wider range of metastability. The constructed network includes 22 entries (Supplementary Data 3) mapped to 509 nodes and 11,912 edges and its illustration is shown in Fig. 2c. Since there is only one target in the net reaction ($Fe_2SiS_4$), higher cutoffs were also chosen for the pathfinding process: $k = 75$ shortest paths and a maximum reaction combination size of 6. The pathfinding process resulted in 340 suggested reaction pathways. However, as before, we demote pathways that include hypothetical compounds or compounds that only exist under conditions far from those considered here. While silicon monosulfide (SiS) has been experimentally synthesized, it primarily appears as a molecular gas and only exists naturally under astrophysical conditions, such as in massive star-forming[33]. In addition, the $SiS_4$ structure was experimentally reported[34] as a delithiated version of $Li_4SiS_4$ but does not appear to have been synthesized on its own. Excluding these two compounds, as well as interdependent reaction steps (see "Methods" section), yields 49 possible pathways. A selection of the top predicted pathways after filtering is shown in Table 3.

The predicted pathways for this synthesis capture many of the intermediates that were experimentally observed. In the experiment, Jiang et al. observed the initial production of $FeS_2$, which was quickly consumed to yield FeSi, $Fe_{1-x}S$, and $SiS_2$; the reaction of these phases yields the target $Fe_2SiS_4$, but not all intermediates were fully consumed in the process. Nearly every predicted path successfully identifies the experimentally observed reaction between $SiS_2$ and FeS, but only 4 of the 49 paths also include the formation of $FeS_2$ and FeSi with this final step. Path 14 is the lowest cost of such paths and involves elemental Si as an additional intermediate phase that reacts with $FeS_2$. While Path 14 is the closest to the experimentally observed pathway, it differs slightly in two regards. In the predicted pathway, $FeS_2$ is consumed by reaction with elemental Si to produce $Fe_2SiS_4$,

and $SiS_2$ is formed from sulfidation of FeSi. In the experiment, FeSi actually appears after $SiS_2$ is formed, suggesting that there is a more complicated missing step (or several steps) involving the reaction of $FeS_2$ with precursors to form a mixture of $Fe_{1-x}S$, FeSi, and $SiS_2$ intermediates.

**Synthesis of $YBa_2Cu_3O_{6.5}$ (YBCO) using barium peroxide.** Miura et al. recently investigated a significant improvement in the solid-state powder synthesis of the superconductor $YBa_2Cu_3O_{6.5}$ (YBCO) by substituting barium carbonate, $BaCO_3$, with barium peroxide, $BaO_2$, resulting in the following net reaction[20]:

$$0.5\ Y_2O_3 + 2\ BaO_2 + 3\ CuO \rightarrow YBa_2Cu_3O_{6.5} + O_2 \quad (6)$$

The reaction network for the Ba–Cu–O–Y chemical system was constructed using a temperature of $T = 1200$ K and 0.1 eV/atom filter. We again selected a less restrictive energy cutoff due to the smaller size of the system, resulting in a network of 54 entries (Supplementary Data 4) mapped to 2973 nodes and 33,957 edges. The network is illustrated in Fig. 2d. Similar to the assisted metathesis systems, the pathfinding was conducted with $k = 20$ paths to each target, a maximum reaction combination size of 5, and an open oxygen chemical potential of $\mu_O = \mu_O^{exp}$(1200 K, 1 atm)[22]. The pathfinding process yielded 52 potential reaction pathways (Supplementary Data 4), only 22 of which do not contain interdependent reaction steps or reaction steps with the hypothetical phases $Cu_8O_7$, $Ba_8Cu_8O_{19}$, and $Y_2Ba_3O_6$. A selection of the top predicted pathways are shown in Table 4.

The predicted reaction pathways to YBCO capture many key aspects of the experimentally observed reaction pathway, but with some slight differences that can be attributed to the inability of the network to capture thermal decomposition/melting. In the experimental pathway, Miura et al. observed that $Ba_2Cu_3O_6$ was

**Table 4 Selection of top predicted reaction pathways to YBa$_2$Cu$_3$O$_{6.5}$ (shown here as Y$_2$Ba$_4$Cu$_6$O$_{13}$).**

| Path | Total cost | Reactions | $\Delta G_{rxn}$ (eV/atom) | Cost |
|---|---|---|---|---|
| 1 | 0.180 | $BaO_2 + CuO \rightarrow BaCuO_2 + 0.5\,O_2$ | −0.160 | 0.177 |
|  |  | $Y_2Ba_4O_7 + 6\,CuO \rightarrow Y_2Ba_4Cu_6O_{13}$ | −0.137 | 0.181 |
|  |  | $3\,Y_2O_3 + 12\,BaCuO_2 \rightarrow Y_2Ba_4O_7 + 2\,Y_2Ba_4Cu_6O_{13}$ | 0.039 | 0.212 |
| 2 | 0.182 | $2\,BaO_2 + 2\,CuO \rightarrow Ba_2Cu_2O_5 + 0.5\,O_2$ | −0.225 | 0.167 |
|  |  | $BaO_2 + CuO \rightarrow BaCuO_2 + 0.5\,O_2$ | −0.160 | 0.177 |
|  |  | $0.8\,Y_4Ba_3O_9 + 5.6\,CuO \rightarrow Y_2Cu_2O_5 + 0.6\,Y_2Ba_4Cu_6O_{13}$ | −0.053 | 0.195 |
|  |  | $3.5\,Y_2O_3 + 9\,BaCuO_2 \rightarrow Y_4Ba_3O_9 + 1.5\,Y_2Ba_4Cu_6O_{13}$ | 0.014 | 0.208 |
|  |  | $2\,Ba_2Cu_2O_5 + Y_2Cu_2O_5 \rightarrow O_2 + Y_2Ba_4Cu_6O_{13}$ | 0.039 | 0.212 |
| 3 | 0.182 | $BaO_2 + CuO \rightarrow BaCuO_2 + 0.5\,O_2$ | −0.160 | 0.177 |
|  |  | $0.8\,Y_4Ba_3O_9 + 5.6\,CuO \rightarrow Y_2Cu_2O_5 + 0.6\,Y_2Ba_4Cu_6O_{13}$ | −0.053 | 0.195 |
|  |  | $Y_2Cu_2O_5 + 4\,BaCuO_2 \rightarrow Y_2Ba_4Cu_6O_{13}$ | −0.010 | 0.203 |
|  |  | $3.5\,Y_2O_3 + 9\,BaCuO_2 \rightarrow Y_4Ba_3O_9 + 1.5\,Y_2Ba_4Cu_6O_{13}$ | 0.014 | 0.208 |
| 4 | 0.185 | $BaO_2 + CuO \rightarrow BaCuO_2 + 0.5\,O_2$ | −0.160 | 0.177 |
|  |  | $0.8\,Y_4Ba_3O_9 + 5.6\,CuO \rightarrow Y_2Cu_2O_5 + 0.6\,Y_2Ba_4Cu_6O_{13}$ | −0.053 | 0.195 |
|  |  | $Y_2Cu_2O_5 + 4\,BaCuO_2 \rightarrow Y_2Ba_4Cu_6O_{13}$ | −0.010 | 0.203 |
|  |  | $2.25\,Ba_2CuO_3 + 2.375\,Y_2O_3 \rightarrow Y_4Ba_3O_9 + 0.375\,Y_2Ba_4Cu_6O_{13}$ | −0.007 | 0.204 |
|  |  | $0.5\,Y_2O_3 + 4\,BaCuO_2 \rightarrow Ba_2CuO_3 + 0.5\,Y_2Ba_4Cu_6O_{13}$ | 0.022 | 0.209 |
| 10 | 0.190 | $4\,BaO_2 + 6\,CuO \rightarrow O_2 + 2\,Ba_2Cu_3O_6$ | −0.231 | 0.166 |
|  |  | $BaO_2 + CuO \rightarrow BaCuO_2 + 0.5\,O_2$ | −0.160 | 0.177 |
|  |  | $Y_2Ba_4O_7 + 6\,CuO \rightarrow Y_2Ba_4Cu_6O_{13}$ | −0.137 | 0.181 |
|  |  | $3\,Y_2O_3 + 12\,BaCuO_2 \rightarrow Y_2Ba_4O_7 + 2\,Y_2Ba_4Cu_6O_{13}$ | 0.039 | 0.212 |
|  |  | $Y_2O_3 + 2\,Ba_2Cu_3O_6 \rightarrow O_2 + Y_2Ba_4Cu_6O_{13}$ | 0.075 | 0.219 |

Seven of the top ten pathways capture the BaCuO$_2$ intermediate, but Ba$_2$Cu$_3$O$_6$ only shows up in Path 10 and beyond. Path 10 represents the closest match to the experimentally observed pathway, in which the predicted Y$_2$Ba$_4$O$_7$ composition appears to facilitate representation of the observed reaction of Y$_2$O$_3$ with Ba–Cu–O melt, which cannot be captured by the reaction network model.

the first intermediate phase to form due to its high thermo-dynamic driving force reacting from the BaO$_2$|CuO$_2$ interface, followed by peritectic decomposition of Ba$_2$Cu$_3$O$_6$ into BaCuO$_2$, subsequent eutectic melting of the BaCuO$_2$|CuO interface, and finally rapid reaction of the Ba–Cu–O melt to form oxygen-deficient YBCO with a final O$_2$ uptake step. In the predicted pathways, we see the formation of BaCuO$_2$ in 7 of the top 10 predicted paths, although Ba$_2$Cu$_3$O$_6$ only appears as an inter-mediate phase beginning with Path 10. Path 10 is the closest to the experimentally observed pathway, passing through both Ba$_2$Cu$_3$O$_6$ and BaCuO$_2$ intermediate phases. However, similar to Paths 1–4, the formation of YBCO from BaCuO$_2$ involves the formation of a Y–Ba–O phase (Y$_2$Ba$_4$O$_7$ or Y$_4$Ba$_3$O$_9$) to balance the reaction; in each case, this intermediate is consumed by CuO to yield more of the YBCO product. Hence the Y–Ba–O intermediate effectively serves as an intermediate composition that allows the network to capture a more complex reaction appearing to involve three components, i.e., the reaction of Y$_2$O$_3$ with a eutectic mixture of BaCuO$_2$–CuO. So while the Y–Ba–O intermediates were not observed in the experiment, their presence in the predicted pathways clearly serves as a convenient mechanism for splitting the Y$_2$O$_3$|Ba–Cu–O(liquid) reaction into two smaller steps that can be captured by the network. Interestingly, two other routes to form YBCO also appear in Paths 2–4, involving the reaction of Ba$_2$Cu$_2$O$_5$ (or BaCuO$_2$) with Y$_2$Cu$_2$O$_5$ to yield YBCO directly. These may serve as alternative routes towards YBCO, which should be explored in future studies.

**Design of synthesis route for novel Mg-ion battery cathode MgMo$_3$(PO$_4$)$_3$O.** Finally, we apply the same reaction network code to demonstrate a different use case: designing synthesis routes to the desired target material. Previously, Rong et al. identified a novel Mg battery cathode material, MgMo$_3$(PO$_4$)$_3$O, which was predicted by DFT calculations to possess an unprece-dented, fast Mg$^{2+}$ mobility in the dilute Mg concentration limit[35]. To our best knowledge, since the publishing of their report in 2017, no experimental studies on the synthesis of

MgMo$_3$(PO$_4$)$_3$O have been published. According to MP, this material is metastable with significantly large energy above the hull of 0.103 eV/atom. Since approximately 20% of all known oxides exhibit energy above hull higher than 100 meV/atom[28], this metric does not directly imply that MgMo$_3$(PO$_4$)$_3$O is impossible to synthesize, but energy above hull of this mag-nitude is typically a sign that the material is at least challenging to synthesize.

To assist in the development of solid-state synthesis routes, we search for possible reactions within a very large reaction network encompassing both the Mg–Mo–P–O system and 39 additional elements that frequently appear in solid-state syntheses, including nearly all the major anions, alkali, and alkaline earth cations, and several transition metals. The total chemical system includes 43 elements: Ag, Al, B, Ba, Be, Bi, Br, C, Ca, Cd, Ce, Cl, Co, Cr, Cs, Cu, F, Fe, I, K, La, Li, Mg, Mn, Mo, N, Na, Ni, O, P, Pb, Rb, S, Sc, Se, Si, Sr, Te, Ti, V, Y, Zn, and Zr. We calculated the thermodynamic stability of all phases in this system at $T =$ 800 K (a relatively low synthesis temperature) and used an energy above hull filter of 0.11 eV/atom, as this cutoff is just large enough to include the target phase. The final reaction network, which includes only reaction edges that can be successfully balanced to form the target phase, contains 21,564 entries mapped to 660,909 nodes and 663,176 edges. We found a total of 2,270 unique reactions yielding MgMo$_3$(PO$_4$)$_3$O as a product.

The full list of 2270 discovered reactions (Supplementary Data 5) includes many reactions that would be challenging to achieve in an experiment for reasons other than the metastability of the target phase, such as reactions that also yield side products that are metastable or highly reactive (e.g., Li metal). One method for distilling down the list of candidate reactions is to use a "metathesis-like" approach and search for reactions that involve the formation of a byproduct that does not include any of the elements in the target, i.e., a second product that does not contain Mg, Mo, P, or O. This is a similar principle as that of metathesis reactions in the sense that it includes the formation of an easily separable compound or highly stable phase which acts as a thermodynamic "sink" that may increase the driving force of the

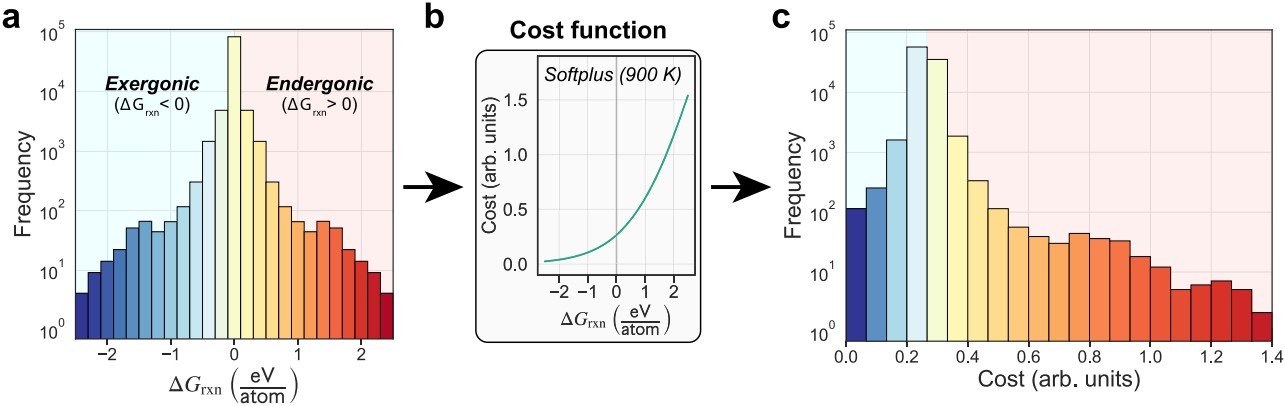

**Fig. 3 Effect of the cost function transformation on reaction energies. a** Distribution of reaction Gibbs free energies (normalized per reactant atom) in the C–Cl–Li–Mn–O–Y chemical system. **b** The softplus cost function, described in Eq. (12), is used to monotonically transform reaction energies. **c** Final distribution of reaction costs (in arb. units) after transformation. Reaction free energy of zero corresponds to a cost of ~0.265.

reaction. Filtering by this constraint reduces the list of candidate reactions down to a more manageable set of 186. Interestingly, nearly every reaction in this candidate set has a similar form: ion exchange from a closely related phase, such as $NaMo_3P_3O_{13}$, $MgFe_3P_3O_{13}$, or $MgV_3P_3O_{13}$, as demonstrated by the example:

$$NaMo_3(PO_4)_3O + MgS_2 \rightarrow MgMo_3(PO_4)_3O + NaS_2$$
$$(\Delta G_{rxn} = 0.044 \text{ eV/atom}) \quad (7)$$

While this reaction may be feasible, it still requires first developing a synthesis route to make the $NaMo_3(PO_4)_3O$ precursor, which is also significantly metastable and not known to be experimentally synthesizable.

Besides ion-exchange routes, there are only two other reactions that emerge. The first is the direct formation of $MgMo_3(PO_4)_3O$ (with no byproduct) from simple ternary phases:

$$Mo(PO_3)_3 + Mg(MoO_2)_2 \rightarrow MgMo_3(PO_4)_3O$$
$$(\Delta G_{rxn} = 0.019 \text{ eV/atom}) \quad (8)$$

While both of these reactants are predicted to be stable by DFT, $Mg(MoO_2)_2$ does not appear to have been previously synthesized in the literature. Assuming this phase is synthesizable (which is likely considering its predicted stability), it is still possible that this reaction might not be spontaneous due to its predicted positive $\Delta G_{rxn}$. However, the small magnitude of this reaction energy is close enough to zero that it may truly be negative, considering both the uncertainties of DFT and the applied Gibbs free energy descriptor.

Finally, the second reaction is an interesting pathway using the ternary nitride, $MgMoN_2$, and $Mo_2P_3O_{13}$:

$$MgMoN_2 + Mo_2P_3O_{13} \rightarrow MgMo_3(PO_4)_3O + N_2$$
$$(\Delta G_{rxn} = -0.054 \text{ eV/atom}) \quad (9)$$

Both precursors of this reaction are predicted to be stable via DFT calculations on the MP database, but we could not find any literature discussing the synthesis of $Mo_2P_3O_{13}$. $MgMoN_2$, on the other hand, has been reported to be synthesized via solid-state reaction[36]. This reaction between ternary nitrides may be a promising synthesis route to $MgMo_3(PO_4)_3O$, assuming that $Mo_2P_3O_{13}$ can first be synthesized —which is a strong possibility given its predicted stability in DFT calculations.

## Discussion

The cost function approach that is central to our model is a necessary transformation for applying pathfinding methods that not only allows for the combination of various reaction metrics (e.g., $\Delta G_{rxn}$, $E_a$) but also serves as a particularly powerful way to navigate uncertainties in thermochemistry data. By transforming reaction free energies to positive costs, we no longer restrict possible reactions to only those with negative free energies (Fig. 3). This seems to be a crucial reason for the model's success since reaction steps that occur experimentally are not always predicted to be thermodynamically favorable using computed—or even experimental—data. For example, the well-studied thermal decomposition of $Li_2CO_3 \rightarrow Li_2O + CO_2$ which appears frequently throughout Table 1 is highly endergonic ($\Delta G_{rxn} > 0$) when modeled with MP data (+0.333 eV/atom at $T = 0$ K). This is somewhat expected, as MP often reports much more negative formation energies for carbonate compounds due to the elemental energy corrections used, which are not always applicable to polyatomic ions. However, this reaction is still highly endergonic using NIST-JANAF experimental data (+0.145 eV/atom at $T = 900$ K), despite the fact that the decomposition of lithium carbonate has been observed to occur spontaneously just above temperatures in this range ($T \sim 900$ K)[37]. We hypothesize that the specific local environment conditions during synthesis, which can differ from the average or global conditions, play a major role in governing the extent of decomposition. Hence it is desirable to retain some degree of flexibility in deciding what is thermodynamically feasible.

It is worth noting that the final ranking of the candidate set of predicted reaction pathways is very sensitive to the choice and form of the cost function. Here, we apply the softplus function to approximate reaction Gibbs free energies derived via a machine-learned model of the vibrational energy that does not explicitly include configurational degrees of freedom. Indeed, in the examples presented, it is not always the lowest total cost path that corresponds to the experimentally observed reaction sequence. The set of suggested pathways also typically includes one or more predictions that are very close to the observed pathways. Since the differences between the highest-ranked (lowest-cost) pathways are small, we strongly recommend users of the model consider several highly ranked pathways. The candidate sets are limited enough in size (often fewer than 100 paths) such that they can be manually inspected by the user.

The primary challenge that is addressed by our work is the high degree of complexity inherent to thermodynamic phase space, which ordinarily leads to a combinatorial explosion during both the creation of the network and subsequent pathfinding steps. For example, consider a reaction network with $N$ phases and a

maximum phase combination size, $n$. If during the graph generation every possible chemical reaction between any two nodes is considered, the number of reactions calculated, $R$, would be:

$$R = \left[ \sum_{i=1}^{n} \binom{N}{i} \right]^2$$
$$= \left[ \binom{N}{1} + \binom{N}{2} + ... + \binom{N}{n} \right]^2 \quad (10)$$

In the C–Cl–Li–Mn–O–Y reaction network shown in Fig. 2a, which contains $N = 76$ distinct phases, the maximum number of possible reactions described by Eq. (10) are $R \approx 5.78 \times 10^3$ ($n = 1$), $8.56 \times 10^6$ ($n = 2$), and $5.36 \times 10^9$ ($n = 3$). This equation scales quickly as constraints are relaxed, often leading to the consideration of millions—or even billions—of possible reactions.

Our implementation reduces the complexity and degrees of freedom of the phase space by introducing a series of filters, including (1) restricting the number of phases considered via thermodynamic stability arguments (energy above the hull), (2) limiting the maximum number of phases present on each side of the reaction to a small number ($n = 2$), (3) using a cost function to prioritize reactions which are more likely to occur, (4) enforcing mass conservation via stoichiometric constraints, and (5) removing interdependent reaction steps. The first two filters work together during graph generation to limit the combinatorial size/complexity of the network. This number can be reduced by decreasing either $N$, $n$, or both. Since it is typically optimal to consider as many phases as possible, and because we see that the complexity scales especially quickly with increasing $n$, it is more favorable to maintain a large $N$ and enforce a constraint of $n = 2$. This choice greatly minimizes the combinatorics of the network but does not inherently sacrifice the complexity of reaction pathways that can occur. In fact, the choice of $n = 2$ may more realistically capture the behavior of reacting solids by dividing reaction pathways into pseudo-elementary steps that more closely follow the free energy surface. This decision is further justified by recent work suggesting that the most thermodynamically favorable pairwise reactions direct the reaction pathway in typical solid-state synthesis procedures[20].

Addressing the combinatorial challenges associated with reaction pathway prediction in large chemical spaces can result in approximations of complex behavior that do not always capture specifics. This complex behavior includes: (1) changes in reaction kinetics due to melting, (2) amorphous intermediates, and (3) defect reactions involving non-stoichiometric compounds. Each of these challenges massively expands the configurational degrees of freedom and necessitates the collection of significantly more thermodynamic data. However, a complete predictive understanding of solid-state synthesis can not be attained without first developing models that incorporate these effects. In future investigations, we expect to address some aspects of these challenges, for example, by including kinetic features leading to more complex reaction pathway selection. While we did not include any kinetic features in this work, we anticipate that several metrics from recent data-oriented studies in chemistry and materials science may be included as extra parameters in the cost function. Such parameters may include the structural (dis)similarity between phases[38,39], the average number of bonds broken/created in a reaction, the change in the information entropy description of atomic configurations, the change in atomic density, etc. The exact weighting of these parameters within the cost function model is unknown as of now, however, and would best be investigated via high-throughput, automated experiments and detailed studies that systematically probe the impact of precursor composition/morphology, reaction conditions, etc. on the resulting reaction pathways.

In conclusion, we designed a solid-state chemical reaction network model constructed from available thermochemistry data and demonstrated the model as a predictor of reaction pathways in solid-state materials synthesis. The framework effectively reduces the large, complex thermodynamic landscape to a computationally tractable structure through (i) creation of a weighted directed graph representation of the available thermodynamic phase space, (ii) mapping of rigorous thermodynamic data and possible heuristics into a versatile cost function, and (iii) application of graph pathfinding algorithms to identify probable reaction routes. While the framework explores reaction trajectories in the most general way possible, allowing for parallel combined pathways, the combinatorial complexity is reduced by chemically motivated filters such as: (1) restricting the number of phases considered via thermodynamic stability arguments, (2) limiting the maximum number of simultaneously reacting phases, and (3) enforcing mass conservation via stoichiometric constraints. As a demonstration, the framework was shown to identify complex reaction pathways comparable to those observed in several experimental solid-state synthesis studies. We envision our methodology to be used to suggest possible synthesis precursors/routes that create efficient thermodynamic conditions for targeting desired phases, as well as identification of byproducts and possible intermediates along synthesis routes. Future work will benefit tremendously by combining the framework 'live' with automated data collection, in situ phase identification, rapid analysis techniques, and automated feedback loops, moving towards active control of solid-state synthesis.

## Methods

**Network model construction.** Figure 4 illustrates the generalized graph structure of a reaction network for any chemical system. Here, the chemical system refers to the set of all $N$ phases $p_i (i = 1, 2, ..., N)$ that can be produced from a designated set of chemical elements. Each reactant/product node on the graph is created by

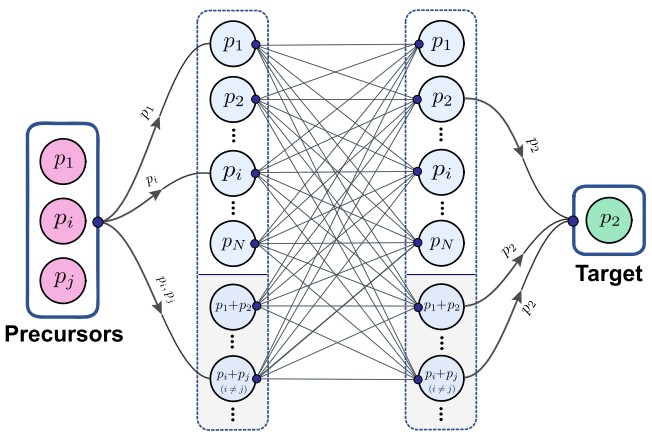

**Fig. 4 Generalized graph architecture of a solid-state chemical reaction network.** The reaction network is constructed for a hypothetical chemical system containing $N$ phases $p_i (i = 1, 2, ...N)$, with nodes made of distinct phase combinations up to maximum size, $n$. The precursors and target nodes link into and out of the densely connected network of reactions, respectively. The edge directions between the reactant and product nodes have been omitted for clarity; an alternate depiction of the network with edge directions is shown in Supplementary Fig. 1. These central edges include chemical reactions weighted by their cost value, as well as zero-weight edges which create loops from product nodes back to reactant nodes so that multiple-step reaction pathways can be captured.

considering combinations of distinct phases up to a maximum size, $n$. This yields the set of all nodes, $P$, given by:

$$P = \{p_i | i \leq N\}$$
$$\cup \{p_i + p_j | i, j \leq N; i \neq j\} \cup \dots \qquad (11)$$
$$\cup \{p_i + p_j + \dots + p_n | i, j, \dots n \leq N; i \neq j \neq \dots \neq n\}$$

In the graph, each of these phase combinations is added twice: first as a reactants node and again as a products node. While higher values of $n$ enable more complex reactions, in general, it suffices to choose $n = 2$ since truly simultaneous reactions among three or more reactants are less likely due to kinetic and steric constraints in a solid composite.

To create the dense set of directed edges at the center of the network, we algorithmically iterate through every possible chemical reaction between all pairs of reactants and product nodes. Using a reaction balancing algorithm, we then solve for the stoichiometric coefficients and add a weighted, directed edge from the reactant node to the corresponding product node for every chemical reaction that is successfully balanced. Note that a vast majority of generated trial reactions cannot be stoichiometrically balanced and hence are excluded from the graph; for example, there are no $x$, $y$, or $z$ that satisfy $xY_2O_3 + yMnO_2 \rightarrow zYMnO_3$, so this reaction edge would not appear in the graph. We also exclude trivial identity-like reactions between identical reactants and products, e.g., $Y_2O_3 \rightarrow Y_2O_3$. The weight of the reaction edge is determined by a "cost function" that maps features of the chemical reaction (e.g., $\Delta\Phi_{rxn}$) to a single cost value. To facilitate product phases being capable of reacting again, zero-weight edges are added which connect each product node to all reactant nodes that contain, as a subset, at least one of the product phases and/or starting reactant phases (regardless of consideration of stoichiometric coefficients). This creates a large degree of cycles in the network that enable the network to capture multiple-step reaction pathways.

Finally, two more nodes are added: one for the synthesis precursors and one for the selected target. These two external nodes act as single-source and destination nodes linking into and out of the dense network of reactions, defining a net (overall) synthesis reaction. The precursors node connects into the network via zero-weight edges directed towards all reactants nodes that contain, as a subset, at least one of the precursor phases. The target node is connected via a set of zero-weight edges directed from all product nodes which contain the target phase.

**Cost function derivation**. The cost function determines the weighting of edges in the network and its nature is critical to the generation of probable reaction pathways. The simplest, and possibly most intuitive, cost function is a one-to-one mapping onto the thermodynamic landscape, such as the measured or calculated Gibbs free energy of reaction, $\Delta G_{rxn}$. However, using unprocessed reaction energies alone poses several problems: (1) negative reaction energies result in infinite cycles during pathfinding, which precludes the use of Dijkstra's algorithm and many other pathfinding methods, (2) kinetic effects and other known heuristics about the reaction are necessarily excluded, and (3) reaction costs are affected by stoichiometric scaling. Instead, here we choose a single, positive cost function that maps the Gibbs free energy of reaction, normalized per reactant atom, to a positive value for each reaction. The choice of a functional mapping also provides the opportunity to create different cost functions for chemical reactions where additional information is known, such as experimental data, kinetic factors, and/or other heuristics. One example of a cost function that captures reaction thermodynamics is the softplus function, which was originally developed for use as an activation function in neural networks[40]. This function maps the Gibbs free energy of a chemical reaction, $\Delta G_{rxn}$, to a positive cost value, $C$, via:

$$C = \ln\left(1 + \frac{273\,\text{K}}{T}e^{\Delta G_{rxn}}\right) \qquad (12)$$

where $T$ is the absolute temperature in Kelvin and $\Delta G_{rxn}$ is the Gibbs free energy of a reaction in units of eV per reactant atom, divided by unity to be dimensionless. Since molar reaction energies scale with the stoichiometric balancing of the reaction, $\Delta G_{rxn}$ must be normalized on a per-atom basis independent of the stoichiometric coefficients. The softplus cost function transforms highly exergonic ($\Delta G_{rxn} \ll 0$) reactions into low (near-zero) cost events, whereas endergonic ($\Delta G_{rxn} > 0$) reactions exhibit a finite cost that smoothly approaches a linear scaling as $\Delta G_{rxn} \rightarrow \infty$. Note that different environmental boundary conditions, such as open elements, can be modeled by replacing $\Delta G_{rxn}$ with $\Delta\Phi_{rxn}$, where $\Phi$ represents a customized thermodynamic potential.

Other monotonically increasing functions, such as those in Fig. 5, were briefly tested. The Rectified Linear Unit (ReLU) function was not appropriate as it did not discriminate between exergonic reactions, which significantly alters the reaction energy distribution and affects the final ranking of pathways. The piecewise linear function preserved the distribution but requires more unknown parameters, and the discontinuity of the first derivative at zero unnecessarily penalized reactions with marginally positive energies. Hence the softplus function was chosen due to its simplicity, smoothness, and differentiable form.

**Application of graph pathfinding methods**. Ideally, predicting a reaction pathway using the network would be fully equivalent to solving the single-source

shortest path problem from graph theory using existing algorithms. However, inorganic chemical reactions rarely trace a set of linear steps even in simple syntheses; instead, the precursor phases often undergo reactions concurrently in parallel or react again. Within the network, parallel reactions can be modeled as simultaneous travel along multiple reaction edges. These reactions must obey mass conservation, and phases produced in one reaction may react with phases in another. These so-called "crossover" reactions, along with the possibility of parallel reactions, prohibit the direct application of shortest path algorithms.

To accommodate parallel paths, we identify not only the single shortest path from precursors to target phase, but a set of $k$-shortest paths to the target that are present in the network. To computationally generate the $k$-shortest paths, we utilize Yen's algorithm[41] which iteratively produces the next $k - 1$ shortest path via deviations from the first shortest path, as calculated with Dijkstra's algorithm[42]. The purpose of using Yen's algorithm is two-fold: (1) to identify a candidate set of reactions that may occur in parallel, and (2) to account for uncertainties in the knowledge of the local synthesis environment, as well as the thermochemistry data used to create the network. For syntheses that involve multiple targets or byproducts (e.g., $CO_2$), pathfinding is performed towards each target phase separately to ensure that all targets appear in the generated set of shortest paths.

To allow for crossover reactions, we first identify all intermediate phases that appear in the $k$-shortest paths to each target and then compute all possible reactions between these intermediates which result in formation of at least one of the targets. These reactions are computed via two approaches: (1) a simple $n$-combinatorial approach analogous to the one used in the network generation, and (2) a compositional phase diagram approach whereby reaction products are predicted to be the set of phases that yield the minimum thermodynamic potential, $\Phi_{min}$, along a compositional tie-line, as previously developed by Richards et al.[26]. This second approach allows crossover reactions to exceed the constraints of the value of $n$ chosen, due to both the inclusion of open elements via a grand potential, as well as the construction of phase diagram simplexes including several product phases in equilibrium. Since it is recommended to select $n = 2$, this second approach makes it possible to additionally capture the reaction of two solids in a flowing gas (e.g., $O_2$), which is a commonly encountered experimental scenario in solid-state synthesis.

**Combining chemical reactions via mass conservation**. When a net reaction is known a priori, reaction steps identified during pathfinding can be linearly combined to satisfy the stoichiometric mass constraints of the overall reaction. These constraints correspond to numerically solving the linear system of equations given by

$$A\mathbf{m} = \mathbf{c} \qquad (13)$$

where $\mathbf{m}$ is a vector containing the multiplicity of each reaction (i.e., the factor by which the entire reaction is multiplied), $A$ is the matrix containing the stoichiometric coefficients of all phases present in all reactions where reactants/products have negative/positive coefficients, respectively, and $\mathbf{c}$ is a vector containing the stoichiometric coefficients of the net synthesis reaction. We solve this system of equations for the multiplicity vector, $\mathbf{m}$, via application of the Moore-Penrose matrix pseudoinverse as implemented within the SciPy package[43].

The total cost, $C_{tot}$, for a balanced reaction pathway with $l$ reaction steps is the weighted mean of individual reaction costs, $s_i$, where the weights are the multiplicities, $m_i$, for each reaction:

$$C_{tot} = \frac{\sum_i^l m_i s_i}{\sum_i^l m_i} \qquad (14)$$

**Identifying interdependent reaction steps**. It is often possible to encounter reaction steps in a reaction pathway that are interdependent, which occurs when two or more reactions exclusively share intermediate phases in such a way that no reaction can proceed without the other(s) happening first. To demonstrate this situation, consider a predicted reaction pathway for the hypothetical net reaction A + B → C + D:

$$A + 0.5\,B \rightarrow D + E$$
$$0.5\,B + F \rightarrow C + G$$
$$E + G \rightarrow F$$

The net stoichiometry of this pathway indeed balances exactly to A + B → C + D, but the second and third reactions are an interdependent pair. In other words, Phase F is a reactant in the second reaction but must be produced by the third reaction, while phase G is a reactant in the third reaction but must be produced in the second reaction. It is often possible to combine interdependent reactions to form a more simple single reaction; in this case, the second and third reactions are combined as $0.5\,B + E \rightarrow C$.

At the end of the reaction pathway search, we consolidate suggested pathways that contain interdependent reactions. This produces more realistic outputs and tends to greatly reduce the number of predicted pathways.

**Thermochemistry data and graph software**. While the chemical reaction network can be created from any thermochemistry data—computed, experimental, or

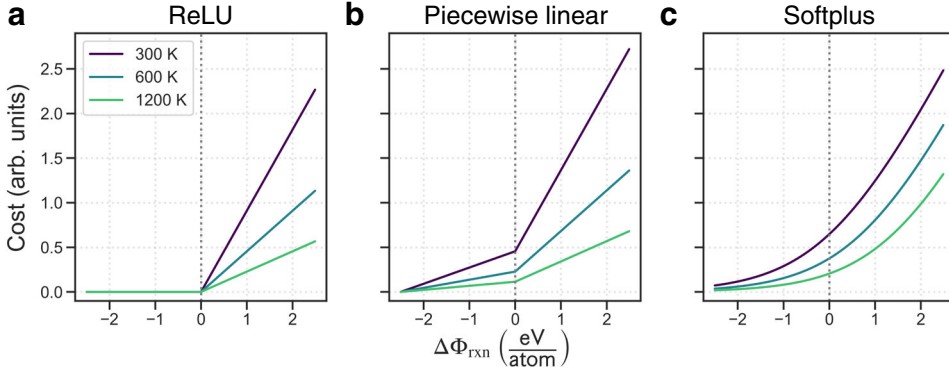

**Fig. 5 Example cost functions.** Each cost function is a monotonically increasing function of reaction free energy, $\Delta\Phi_{rxn}$, and scaled according to temperature, $T$. The functions include (**a**) Rectified Linear Unit (ReLU), **b** piecewise linear, and (**c**) softplus. Each has the result that the thermodynamic free energy of the reaction is rectified to positive values and scaled such that lower reaction temperatures result in higher costs.

a combination of both— in this work, we primarily employ the Materials Project database, which contains well-benchmarked ab initio calculated formation enthalpies for over one hundred thousand different materials as calculated with density functional theory (DFT)[21,44]. To capture the temperature dependence of vibrational degrees of freedom, we employ the machine-learned Gibbs free energy descriptor reported by Bartel et al.[45], which estimates the finite temperature contribution to the Gibbs free energy of formation of solids, $\Delta G_f(T)$. This contribution incorporates both temperature-dependent enthalpic and entropic effects, although the entropic contribution ($TS$) typically dominates. The elemental Gibbs free energies used for these formation energy calculations are acquired from FactSage[22]. The Gibbs free energies of formation for non-elemental gases (e.g., $CO_2$), as well as a small set of solid compounds ($A_2O$ and $A_2CO_3$; $A$=Li, Na) are acquired from NIST-JANAF experimental thermochemical tables[23].

Thermodynamic phase diagram calculations and reaction balancing procedures are performed using algorithms implemented in the pymatgen package[46]. Graphs are implemented using the graph-tool package[47] and visualized in Fig. 2 using Graphistry Hub[48].

## Data availability

All results data supporting the findings of this paper are included within the article or supplied as Supplementary Data. Ab initio thermochemistry data used in this work are available via the open-access Materials Project database[21] (version 2020.09.08). Supplemental experimental data were acquired from the open-access NIST-JANAF database[23] and are also included as part of the provided software. Source data for Fig. 3 are provided with the paper. Source data are provided with this paper.

## Code availability

A Python implementation of the solid-state reaction network method is available at https://github.com/GENESIS-EFRC/reaction-network, archived in ref. [49], and provided as Supplementary Software 1.

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

## Acknowledgements

This work was supported as part of GENESIS: A Next-Generation Synthesis Center, an Energy Frontier Research Center funded by the U.S. Department of Energy, Office of Science, Basic Energy Sciences under Award Number DE-SC0019212. This research used resources of the National Energy Research Scientific Computing Center (NERSC), a U.S. Department of Energy Office of Science User Facility operated under Contract No. DE-AC02-05CH11231. The authors would like to thank J. Neilson, P. Todd, O. Kononova, and C. Bartel for their helpful discussion regarding the reaction network model, as well as E. Persson for math skills, and L. Meyerovich for assistance with graph visualization.

## Author contributions

M.J.M. and S.S.D. conceived the idea of the presented work. M.J.M. designed and developed the code with feedback from S.S.D. and K.A.P. M.J.M. performed all calculations and wrote the manuscript with the guidance of S.S.D. and K.A.P.

## Competing interests

The authors declare no competing interests.
