## [Peer Review File · Nature Communications]

REVIEWER COMMENTS

Reviewer #2 (Remarks to the Author):

The paper under review presents a novel method for representing solid state reactions as a graph and finding minimum cost traversals that may represent potential pathways for materials synthesis. I enjoyed reading the manuscript and found the presentation, both in terms of the writeup and figures to be well polished. I didn't find any grammatical or typographical errors, the figure captions do a good job of explaining the figures, and the figures themselves are nicely prepared.

My main concerns about publishing this work in the present venue are about impact and the substantivity of the work described. To me, the writing included in the Results section reads much more like "Methods", while the actual results appear limited to the experimental result for the YMnO_3 system, described in the third subsection of the Methods. While it is certainly an interesting result that this algorithmic method can reproduce the theoretical and experimental results in MatDB, this result is still limited only to this one system and cannot be generalized to others. In the conclusion, there is a list of future work which includes experimentation -- I agree that evaluating the usefulness of this technique, either with new experiments, or a greater number of material systems from open datasets (perhaps including data beyond Materials Project, as well), is essential to understand the practical value and efficacy of the proposed method. Without this, the paper reads very methods-heavy (though they are certainly interesting methods) with underdeveloped results.

At the very least, if this were to be published in its current form, I'd recommend restructuring the article to include the methods writeup in its section and to appreciably expand the results section with greater detail on the demonstration for YMnO_3 , including perhaps some of the content in the supplementary information.

Reviewer #3 (Remarks to the Author):

The authors proposed one of a few study for inorganic reaction pathway prediction, which is very interesting and pioneering. The shortest path approach is heuristic but is sufficient as early attempts. However, the validation of this approach is not solid and should be improved. To demonstrate the universality, the authors should do hold-out evaluation to report if the algorithm can rediscover several non-trivial well-known experimentally verified reaction pathways.

In the case study of YMnO_3 , two reaction pathways are reported. The first one involves a metastable compound. The second ranked pathway by cost, also only closely matches the experimentally reported assisted metathesis pathway.

This pathway is nearly identical to the experimentally reported pathway;

==

what is the reason that the proposed algorithm fails to find the identical experimental pathway?

we identify not only the single shortest path from precursors to target phase, but all k-shortest paths.

==

In this case, what is the motivation of picking k-path with shortest ones?

Malik, Shreshth A., Rhys EA Goodall, and Alpha A. Lee. "Materials Graph Transformer predicts the outcomes of inorganic reactions with reliable uncertainties." arXiv preprint arXiv:2007.15752 (2020).

==

Can you discuss the similarity and difference of your paper with this one?

page 6:

Of the total 195,708 pathways considered, only two combined reaction pathways satisfy the stoichiometric constraints of the net reaction

will this cause the issue that for some materials, the proposed approach cannot find any pathway for it? or some constraints should be relaxed to find at least a path?

page8:

The use of filters seems to also have risk of losing feasible search spaces. Can you discuss why it won't rule out the most suitable pathways that we aim to find?

have you consider to use some transformer deep neural network model to address the huge design space issue?

Schwaller, Philippe, Teodoro Laino, Théophile Gaudin, Peter Bolgar, Christopher A. Hunter, Costas Bekas, and Alpha A. Lee. "Molecular transformer: A model for uncertainty-calibrated chemical reaction prediction." ACS central science 5, no. 9 (2019): 1572-1583.

page9:

some discussion on the technical differences of reaction prediction of inorganic materials and organic molecules are needed in the introduction part or the discussion part.

Neglecting the large literature on organic reaction learning is not appropriate.

Do the authors have the plan to release the graphs of the network so that other machine learning approaches rather than the shortest path approach can be derived?

Reviewer #4 (Remarks to the Author):

This paper proposes a framework for predicting and suggesting solid-state inorganic reaction pathways in order to perform inorganic synthesis by design. The model proposed is a chemical reaction network constructed from the thermochemistry databases. Path finding algorithms are used to suggest possible reaction pathways for synthesis of required solid-state materials. The applicability of the model is demonstrated by predicting a complex metathesis reaction pathway toward yttrium manganese oxide (YMnO₃). The paper is quite well-written for the most part.

I have the following comments/questions/suggestions for possible improvement:

- It is not clear if this is a "prediction" problem, as indicated in the title and paper. As far as I can tell, all the reactions in the Supplementary Tables 2-4 were present in the chemical reaction network (with pre-calculated Gibbs energy and the derived cost), and path-"finding" algorithms were used to identify low cost paths that satisfy the stoichiometric constraints of "a net/overall reaction which is known a priori". To me, it appears to be more like a search problem. I think prediction would have been if the model was able to predict new unseen reactions (without underlying DFT calculations).

- The graph search problem which the work seems to actually address is not trivial though, and involves significantly complex constraints, which indeed seem to be handled very well by the proposed work. One of the keys in enabling that is the construction of the chemical reaction

network, which is well-described for the most part, but in Figure 2, "the edge directions between the reactant and product nodes have been omitted for clarity". Since there can be edges in both directions, the authors may want to explain this further with an example, perhaps with a separate figure if needed.

- The derivation of the cost function as a function of Gibbs energy is interesting. I would have liked to see some more discussion about the authors' view on its suitability for use as a cost function. For example, is it additive? Does it physically represent anything in this context?

- Towards the end of the discussion section, the authors claim: "...the network can also be used to identify shortest paths to or from any nodes in the network." I am not sure if enough evidence has been presented to substantiate that claim, as the results are primarily focused at identifying an existing reaction pathway in a single system with all parameters (precursors, target, temperature) informed by the experiment.

- Overall the work is technically strong and the preliminary study on YMnO_3 is indeed compelling. While this may be sufficient for a relatively lower ranked journal, I believe for Nat Comm, it may be useful to provide additional evidence for the generalizability of the proposed method. For example, the authors may consider "suggesting" some unexplored-yet reaction pathways to synthesize some interesting compounds. The authors' ideas of "speculating possible likely products when no net reaction information is known" and "identify promising precursors which yield efficient chemical routes towards desired targets" are also very interesting. I understand that the authors may not want to put everything in one paper, but demonstrating better generalizability in some of the above ways could possibly help make the paper stronger and more complete.

- The authors provide code via GitHub, which is great. It is not clear though if the released code and underlying data (reaction network) is limited to the YMnO_3 example, or is derived more broadly from the entire Materials Project.

- Minor: Occasional typos:

-- Page 4: "...from each from each..."

-- Page 6: "...the use of Yen's algorithm to generate many such low-cost paths narrows down the range of probable reactions..." It seems that it actually increases the set of candidate paths, which is totally desirable, as explained by the authors themselves.

We are very thankful to the reviewers, whose thoughtful comments and suggestions have greatly helped us improve the manuscript. We have updated the manuscript to include additional results that we believe provide a better indication of the generalizability of the proposed reaction network method.

The updated Results section now includes reaction network analysis of **three additional** experimentally-studied synthesis procedures (along with the original YMnO_3 example), as well as an **additional demonstration** of the network as a tool for devising synthesis routes to a predicted battery cathode material that has not yet been synthesized. With this revision, we have also moved much of the original text discussing the network methods to the formal Methods section of the paper, and updated the Discussion to reflect the new results.

Please note that while the methods used in the revised manuscript are nearly identical to those used in the original manuscript, we did make two minor additions to our code that (positively) impacted the results: 1) an improved crossover reaction step that now incorporates knowledge of the compositional phase diagram to allow for both open elements and reactions with more than two products, and 2) better ranking/filtering on the final predicted pathways, including the removal of pathways with “interdependent” reaction steps.

Our point-by-point response to the reviewers’ comments can be found below. The reviewers’ original comments are in *black italics*, and our responses in **blue**. Quoted changes are shown in **highlighted yellow**. We have also included a version of the manuscript with tracked changes. Once again, we thank the reviewers for their positive feedback and constructive comments.

Reviewer #2 (Remarks to the Author):

The paper under review presents a novel method for representing solid state reactions as a graph and finding minimum cost traversals that may represent potential pathways for materials synthesis. I enjoyed reading the manuscript and found the presentation, both in terms of the writeup and figures to be well polished. I didn't find any grammatical or typographical errors, the figure captions do a good job of explaining the figures, and the figures themselves are nicely prepared.

We thank the reviewer for their positive comments!

My main concerns about publishing this work in the present venue are about impact and the substantivity of the work described. To me, the writing included in the Results section reads much more like "Methods", while the actual results appear limited to the experimental result for the YMnO_3 system, described in the third subsection of the Methods.

The reviewer makes a great suggestion for improving the readability of the paper, which we have addressed in the revised manuscript. We restructured the text to move all content discussing method-related specifics from the Results section to the Methods section. We thank the reviewer again for this suggestion.

While it is certainly an interesting result that this algorithmic method can reproduce the theoretical and experimental results in MatDB, this result is still limited only to this one system and cannot be generalized to others. In the conclusion, there is a list of future work which includes experimentation -- I agree that evaluating the usefulness of this technique, either with new experiments, or a greater number of material systems from open datasets (perhaps including data beyond Materials Project, as well), is essential to understand the practical value and efficacy of the proposed method. Without this, the paper reads very methods-heavy (though they are certainly interesting methods) with underdeveloped results.

This is a valid concern raised by the reviewer and we believe that the additional results added now address the generalizability outside of the original YMnO_3 example. While there is only a small number of previous experimental studies that dissect solid-state reaction pathways, the initial successes we present show that our methods should be helpful to analyze and better understand future syntheses using solid-state methods. Future work and current challenges to the model (e.g., amorphous phases, melting) are discussed and solving these would certainly help improve predictions, and will be the topic of future improvements to the methodology. However, we believe that our results demonstrate the ability of our current methodology to capture non-trivial behavior of several experimentally verified reaction pathways.

Furthermore, our methods are not limited to Materials Project (MP) data and can be applied to any general thermochemistry dataset that contains entries with a defined 1) composition and 2) formation energy. While much of our data comes from MP due to its size, breadth, and ease of access, we do include experimental data from NIST-JANAF thermochemistry tables for gases, as well as few solid compounds like Li_2CO_3 and Li_2O , showing that it is indeed possible to use any thermochemistry dataset provided that the complete set of data covers a good portion of the energy-composition chemical landscape of interest.

At the very least, if this were to be published in its current form, I'd recommend restructuring the article to include the methods writeup in its section and to appreciably expand the results section with greater detail on the demonstration for YMnO_3 , including perhaps some of the content in the supplementary information.

We thank the reviewer again for their constructive feedback. As previously mentioned, we restructured the manuscript by transferring methods-related content to the Methods section, expanding the Results, and updating the Discussion section accordingly.

Reviewer #3 (Remarks to the Author):

The authors proposed one of a few study for inorganic reaction pathway prediction, which is very interesting and pioneering. The shortest path approach is heuristic but is sufficient as early attempts. However, the validation of this approach is not solid and should be improved. To demonstrate the universality, the authors should do hold-out evaluation to report if the algorithm can rediscover several non-trivial well-known experimentally verified reaction pathways.

We thank the reviewer for their constructive comments on validation. As previously mentioned, the additional results in the revised manuscript significantly improve upon the validation of our approach and

demonstrate the ability of our methods to capture non-trivial behavior of several experimentally verified reaction pathways. For example, in the Li-based assisted metathesis example, the network successfully captures both the experimentally proposed (kinetically favorable) low-temperature pathway for formation of YMnO_3 from reaction of YOCl and LiMnO_2 , as well as the alternate pathway reacting Mn_2O_3 and Y_2O_3 that was shown to simultaneously occur in parallel at sufficiently high temperatures. In the Na-based system, the particular progression of the Na_xMnO_2 intermediate was not directly captured, but the formation of Na-Mn-O intermediates and reaction of Y_2O_3 and an MnO_2 species match what was experimentally observed. In the Fe_2SiS_4 example, one of the top pathways captures all four experimentally observed intermediates (FeS_2 , SiS_2 , FeSi , and FeS) as well as the final reaction step ($\text{SiS}_2 + \text{FeS} \rightarrow \text{Fe}_2\text{SiS}_4$). Finally, in the YBCO example, the network captures the intermediates BaCuO_2 and $\text{Ba}_2\text{Cu}_3\text{O}_6$ -- and while it does not capture the decomposition/melting aspects of the reaction, it is apparent from the suggested pathway that one of the intermediates facilitates representation of the reaction from melt. Specifically, this intermediate appears in two steps, which if combined, yields a reaction between 3 phases -- one that would not be likely without facilitation via melting.

While we agree that hold-out validation is desirable for any model used to make predictions from training data, we do not believe it is feasible for the construction of this type of network model. To clarify, our model is not actually “trained” on any data, but rather constructed by directly operating on thermodynamic data. In effect, our construction holds-out all validation data. Due to the very small amount of published experimental solid state reaction pathway data -- which we estimate to be just a few dozen studies -- our best attempts to validate the model are via case-by-case comparisons, as is now done in the manuscript for four case studies. In the future, as automated experimentation and in-situ characterization becomes more commonplace, it should be possible to incorporate more of a true machine learning approach.

In the case study of YMnO_3 , two reaction pathways are reported. The first one involves a metastable compound. The second ranked pathway by cost, also only closely matches the experimentally reported assisted metathesis pathway.

This pathway is nearly identical to the experimentally reported pathway;

==

what is the reason that the proposed algorithm fails to find the identical experimental pathway?

In the original manuscript, the methods did not identify the exact experimental pathway as reported in the Todd et al. (2019) paper, because the reported pathway featured a reaction step with three products, which could not be captured by our previous version of the network. There was also no indication of the oxidation of $\text{YOCl} \rightarrow \text{Y}_3\text{O}_4\text{Cl} \rightarrow \text{Y}_2\text{O}_3$. This missing oxidation was partially due to the lack of inclusion of $\text{Y}_3\text{O}_4\text{Cl}$ in the data set (this phase is not present in MP), as well as the limitation to a maximum size of 4 reactions in the final produced pathways.

In the revised manuscript, however, we are able to capture the experimentally reported pathway by enabling the inclusion of reactions with 3 or more products. We accomplished this, while avoiding a combinatorial explosion, by using the existing phase diagram code to predict the thermodynamically favorable set of products. We also increased the maximum reaction combination size to 5 (less stringent

cutoff), which allowed us to capture more complex pathways. Furthermore, we were able to see signs of the $\text{YOCl} \rightarrow \text{Y}_3\text{O}_4\text{Cl} \rightarrow \text{Y}_2\text{O}_3$ oxidation that was experimentally reported, both because of the larger combination size used and because we added a calculated entry for $\text{Y}_3\text{O}_4\text{Cl}$ via the same DFT methods and corrections used in the MP database. Overall, these improvements in reaction pathway construction are mainly a result of more computationally efficient methods, which suggest that further algorithmic scaling would enable investigating more complex reaction behavior.

we identify not only the single shortest path from precursors to target phase, but all k-shortest paths.

==

In this case, what is the motivation of picking k-path with shortest ones?

The motivation of using the k-shortest paths algorithm is to produce a candidate set of the most likely reaction steps which may be encountered during the synthesis. Importantly, this candidate set is just small enough to avoid a combinatorial explosion of possible paths through the network, while also providing flexibility to uncertainty in the data used to construct the network. In other words, for the methods here to be tractable we need to find a careful balance between considering all possible reaction steps (which there are often 10-100K in a network), and trusting that our data and cost representations are accurate. If our cost representation of the reactions accurately reflects experimental conditions (correct reaction energies, compositions, etc.) then we should be able to decrease the size of the set of reactions considered. It should also be noted, however, that producing a set of reactions also allows us to consider reactions with very similar costs which may actually occur in parallel, and thus would not be capturable in a single shortest path representation.

Malik, Shreshth A., Rhys EA Goodall, and Alpha A. Lee. "Materials Graph Transformer predicts the outcomes of inorganic reactions with reliable uncertainties." arXiv preprint arXiv:2007.15752 (2020).

==

Can you discuss the similarity and difference of your paper with this one?

In the suggested paper, the authors utilize the graph architecture to create a fixed-length representation of an experimental solid-state synthesis procedure with the goal of predicting the stoichiometry of the major target. While our paper also utilizes a graph structure to encode connectivity between phases, the goal of our network construction is explicitly to derive the many possible smaller reaction steps, as well as discrete intermediate phases, which make up the overall reaction pathway of a synthesis procedure. More simply stated, in addition to being able to predict the most likely target or optimal precursors, our network was designed to predict the steps in between. As mentioned, due to the very small amount of in situ experimental reaction pathway data, this type of prediction would not be successful via a machine learning approach. Instead, we demonstrate that the reaction graph can be constructed and pathway steps extracted using a few physically-motivated principles (e.g., cost transformation, mass balance).

page 6:

Of the total 195,708 pathways considered, only two combined reaction pathway satisfy the stoichiometric constraints of the net reaction

will this cause the issue that for some materials, the proposed approach cannot find any pathway for it? or some constraints should be relaxed to find at least a path?

Yes, it is possible that the proposed approach may not identify a pathway after applying mass conservation constraints, however this was not encountered in any of the presented results. We hypothesize that the primary reasons this might happen would be either due to highly restrictive parameters, (i.e., not enough phases, reactions, or combinations considered) or because the experimental pathway mostly passes through non-stoichiometric compounds and/or amorphous/liquid phases, which may exist as solid solutions across a composition range. While the mass conservation constraint could be loosened to allow for other pathways to be produced, it is not clear that this would produce logical or useful predictions. Thankfully, as discussed in the manuscript in the YBCO example, it is still possible for the methods to produce pathway predictions that encompass the net effect of a melted/amorphous phase, often via “filler” compositions, like what we demonstrated with the $Y_2Ba_4O_7$ composition in that example.

page8:

The use of filters seems to also have risk of losing feasible search spaces. Can you discuss why it won't rule out the most suitable pathways that we aim to find?

Using a filter-based approach does indeed run the risk of excluding phases, reactions, and reaction pathways that we aim to find. However, the initial work here suggests that the results are less impacted by the specific values used for the filters (provided they are reasonable), and more impacted by major deficiencies in the data set used. In particular, the lack of certain compositions necessarily prevents identification of a reaction pathway that includes any unknown composition. This was the main reason a partial pathway was missing in the original submission of the manuscript -- Y_3O_4Cl was not included in the original MP database, so pathways through this composition were not discoverable. Furthermore, phases with off-stoichiometry (e.g., Na_xMnO_2) are only captured for specific values of x , so complex defect-based reactions are not easily captured, which may prevent finding some suitable pathways. While we do not yet have a solution for off-stoichiometric compositions, missing phases can be accommodated by performing substitution-based structure predictions prior to building the reaction network. In other words, it is possible to fill out missing compositions (and their structures) within chemical spaces by ionic substitution in existing structures, as outlined in the following paper:

Hautier, G., Fischer, C., Ehlacher, V., Jain, A., & Ceder, G. (2011). Data Mined Ionic Substitutions for the Discovery of New Compounds. *Inorg. Chem*, 50, 656–663. <https://doi.org/10.1021/ic102031h>

This was how the added structure for Y_3O_4Cl was generated in our manuscript, and is also how many hypothetical materials are added to the Materials Project. Since this still requires knowing starting structural frameworks in similar chemical systems, it is also possible to do a less constrained exploratory approach. Trial compositions can be generated through machine learning-based combinatorial screening approaches, such as the one presented here:

Meredig, B., Agrawal, A., Kirklin, S., Saal, J. E., Doak, J. W., Thompson, A., ... Wolverton, C. (2014). Combinatorial screening for new materials in unconstrained composition space with machine learning. *PHYSICAL REVIEW B*, 89, 94104. <https://doi.org/10.1103/PhysRevB.89.094104>

And then the structures of these compositions can be determined through any standard crystal structure prediction approach, such as the random superlattice approach presented here:

Stevanović, V. (2016). Sampling Polymorphs of Ionic Solids using Random Superlattices. *Physical Review Letters*, 116(7), 075503. <https://doi.org/10.1103/PhysRevLett.116.075503>

Have you consider to use some transformer deep neural network model to adress the huge design space issue?

Schwaller, Philippe, Teodoro Laino, Théophile Gaudin, Peter Bolgar, Christopher A. Hunter, Costas Bekas, and Alpha A. Lee. "Molecular transformer: A model for uncertainty-calibrated chemical reaction prediction." *ACS central science* 5, no. 9 (2019): 1572-1583.

We speculate that a transformer model to predict the formation energy for new phases would assist in filling out the phase diagram and potentially prioritize phases for computation. However, since databases such as MP are the primary source of training data for ML models, it's very likely that such transformer models will be biased in a similar manner.

page9:

some discussion on the technical differences of reation prediction of inorganic materials and organic molecules are needed in the introduction part or the discussion part.

Neglecting the large literature on organic reaction learning is not appropriate.

This is a great suggestion by the reviewer, and we have added additional text (and references) in the Introduction which mentions reaction pathway prediction for organic vs. inorganic syntheses.

Line 57: Explicit modeling, as well as reaction network models derived from the atomistic potential energy surface (PES), have been successful in predicting chemical reaction pathways in molecular systems^{14,15} but are much less developed for solid-state *periodic* systems, where monitoring each atom's coordinates and interactions over the large time and spatial scales necessary rapidly becomes intractable.

Line 77: Recent work also suggests that the computational prediction of reaction pathways in ceramic powder-based synthesis does not always require atomistic methods; significant predictive power can be derived from local thermodynamic equilibrium calculations of pairwise solid-solid interfaces.

Do the authors have the plan to release the graphs of the network so that other machine learning approaches rather than the shortest path approach can be derived?

Definitely! All code used to create the graph networks is accessible via Github (<https://github.com/GENESIS-EFRC/reaction-network>). While we share all pathway-related raw data in the Supplementary Data, we will not upload the network objects themselves (which are between 1-10 GB range even when compressed), both since they are reasonably easy to obtain for the examples in the paper and since the time required to load the graph into memory is comparable to generating it from scratch. In fact, the code used to create the graph objects is parallelized via Dask and can typically run in a few minutes even on a laptop computer.

We suggest referencing the demonstration Jupyter notebook (*demo.ipynb*) on the Github repository, which walks through recreating all networks and pathway analysis presented in the manuscript.

Reviewer #4 (Remarks to the Author):

This paper proposes a framework for predicting and suggesting solid-state inorganic reaction pathways in order to perform inorganic synthesis by design. The model proposed is a chemical reaction network constructed from the thermochemistry databases. Path finding algorithms are used to suggest possible reaction pathways for synthesis of required solid-state materials. The applicability of the model is demonstrated by predicting a complex metathesis reaction pathway toward yttrium manganese oxide (YMnO₃). The paper is quite well-written for the most part.

We thank the reviewer for their positive comments!

I have the following comments/questions/suggestions for possible improvement:

- It is not clear if this is a "prediction" problem, as indicated in the title and paper. As far as I can tell, all the reactions in the Supplementary Tables 2-4 were present in the chemical reaction network (with pre-calculated Gibbs energy and the derived cost), and path-"finding" algorithms were used to identify low cost paths that satisfy the stoichiometric constraints of "a net/overall reaction which is known a priori". To me, it appears to be more like a search problem. I think prediction would have been if the model was able to predict new unseen reactions (without underlying DFT calculations).

We are grateful for the reviewer pointing out that the original manuscript is unclear as to why our proposed method is predictive rather than simply search-based. We believe that our model does indeed create predictions for reaction pathway steps, because the reactions and associated pathways themselves are not previously derived in any thermochemical database. During the graph construction, we identify and actually determine these possible reactions ourselves by using only the knowledge of possible phases (composition/structure + formation energy pairs). We then calculate the free energy of the reaction, including contribution from a previous model for approximating the transformation to temperature-dependent formation energies from DFT-computed data. This is all done while constructing the layout of the network.

The model can predict new/unseen reactions in the sense that all reaction edges will be created in the network even for additional, hypothetical phases. While we only use DFT-based (with auxiliary experimental entropic data) phase data in our examples, it is possible to include new compositions and estimates of their free energy of formation (e.g. via linear interpolation of energy from the phase diagram), and this feature has already been added to the code. In other words, the network is not dependent on using precomputed reactions, or even precomputed data.

- The graph search problem which the work seems to actually address is not trivial though, and involves significantly complex constraints, which indeed seem to be handled very well by the proposed work. One

of the keys in enabling that is the construction of the chemical reaction network, which is well-described for the most part, but in Figure 2, "the edge directions between the reactant and product nodes have been omitted for clarity". Since there can be edges in both directions, the authors may want to explain this further with an example, perhaps with a separate figure if needed.

This is indeed a confusing point which we have addressed in the revised manuscript by adding Supplementary Figure 1, which uses an alternative layout of the network to show the edges which "loop back" from the products to the reactants (going in reverse direction to the reaction edges). We have decided to keep the original schematic figure in the manuscript due to its better clarity in other aspects.

Supplementary Figure 1: Alternate drawing of reaction network model, illustrating reaction pathways and simple loops. The cost/weight of edges are shaded by the colors red (high-cost), green (low-cost), and gray (zero-cost). This drawing is for a hypothetical chemical system containing just eight phases (labeled by Greek letters). In this system, only two reaction pathways exist to the target, δ , from the precursors α , β , and γ . The first pathway is the simple one-step reaction: $\alpha + \beta \rightarrow \delta + \epsilon$. The second pathway contains two steps and features intermediate phases: $\beta + \gamma \rightarrow \eta + \theta \rightarrow \delta + \zeta$. In this diagram, only simple loops are illustrated; however, the network features many more zero-cost looping edges between a product node and any reactant node which contains a subset of the product phases and/or the precursor phases.

- The derivation of the cost function as a function of Gibbs energy is interesting. I would have liked to see some more discussion about the authors' view on its suitability for use as a cost function. For example, is it additive? Does it physically represent anything in this context?

We have expanded our comparison of the cost functions in the Methods section, and enhanced discussion of its implications in the Discussion section. Furthermore, we have moved the cost function figure from the Supplementary Information to the main body of the manuscript (Figure 5). The additional equation (14) of Methods describes how the reaction pathway cost is calculated. The following added paragraph briefly discusses the suitability of the softplus function:

Line 779: Other monotonically increasing functions, such as those in Fig. 5, were briefly tested. The Rectified Linear Unit (ReLU) function was not appropriate as it did not discriminate between exergonic reactions, which significantly alters the reaction energy distribution and affects the final ranking of pathways. The piecewise linear function preserved the distribution but requires more unknown parameters, and the discontinuity of the first derivative at zero unnecessarily penalized reactions with marginally positive energies. Hence the softplus function was chosen due to its simplicity, smoothness, and differentiable form.

- Towards the end of the discussion section, the authors claim: "...the network can also be used to identify shortest paths to or from any nodes in the network." I am not sure if enough evidence has been presented to substantiate that claim, as the results are primarily focused at identifying an existing reaction pathway in a single system with all parameters (precursors, target, temperature) informed by the experiment.

Note that the generalizability of pathfinding methods does allow them to be used from any node to any other node in the network, however the applied physical constraints and postprocessing steps may change if less is known about the synthesis parameters. The paragraph which originally included the referenced sentence (and other speculative applications) has been removed, and instead we have added a demonstration of the network being used in a less constrained way to predict new synthesis routes to a target material, $\text{MgMo}_3(\text{PO}_4)_3\text{O}$, without any prior knowledge of precursors.

- Overall the work is technically strong and the preliminary study on YMnO_3 is indeed compelling. While this may be sufficient for a relatively lower ranked journal, I believe for Nat Comm, it may be useful to provide additional evidence for the generalizability of the proposed method. For example, the authors may consider "suggesting" some unexplored-yet reaction pathways to synthesize some interesting compounds. The authors' ideas of "speculating possible likely products when no net reaction information is known" and "identify promising precursors which yield efficient chemical routes towards desired targets" are also very interesting. I understand that the authors may not want to put everything in one paper, but demonstrating better generalizability in some of the above ways could possibly help make the paper stronger and more complete.

We greatly appreciate the reviewer's suggestion to better demonstrate the generalizability of the proposed method. While we are currently in the process of completing follow-up works to address some of these ideas, we did decide to include in our updated Results and example of using the code to "identify promising precursors which yield efficient chemical routes towards desired products", in this case towards the previously unsynthesized Mg battery cathode material $\text{MgMo}_3(\text{PO}_4)_3\text{O}$.

- The authors provide code via GitHub, which is great. It is not clear though if the released code and underlying data (reaction network) is limited to the YMnO_3 example, or is derived more broadly from the entire Materials Project.

The GitHub repository includes all code needed both to fully reproduce the results in the manuscript, as well as to apply the methods to any other desired system. The GitHub also includes an example Jupyter notebook (demo.ipynb) which serves as a walkthrough of how we generated and analyzed our results.

We have also updated the wording of our Code Availability statement to make this more clear, thanks to the reviewer's comment.

Line 924: All code used in this work is available as an installable Python package located in the reaction-network repository on GitHub, found at <https://github.com/GENESIS-EFRC/reaction-network>. An included Jupyter notebook (demo.ipynb) serves as a walkthrough to recreate the results and analysis presented in the manuscript and can be extended to model any system of the user's choice.

- *Minor: Occasional typos:*

-- Page 4: "...from each from each..."

-- Page 6: "...the use of Yen's algorithm to generate many such low-cost paths narrows down the range of probable reactions..." It seems that it actually increases the set of candidate paths, which is totally desirable, as explained by the authors themselves.

Thank you to the reviewer for pointing out these typos; they have been addressed in the revised manuscript.

With these revisions, we hope that you will find our manuscript acceptable for publication.

REVIEWERS' COMMENTS

Reviewer #2 (Remarks to the Author):

Thank you for your extensive and thorough revisions. I find the changes entirely adequate and compelling and see no further issue to prevent publication.

Reviewer #3 (Remarks to the Author):

The revised manuscript has addressed all my concerns over the paper especially for the validation part with more case studies.

Reviewer #4 (Remarks to the Author):

The authors have extensively revised the manuscript to address my comments.

Reviewer #2 (Remarks to the Author):

Thank you for your extensive and thorough revisions. I find the changes entirely adequate and compelling and see no further issue to prevent publication.

We thank the reviewer for their helpful suggestions and support for publication.

Reviewer #3 (Remarks to the Author):

The revised manuscript has addressed all my concerns over the paper especially for the validation part with more case studies.

We thank the reviewer for their helpful comments and suggestions to improve the validation.

Reviewer #4 (Remarks to the Author):

The authors have extensively revised the manuscript to address my comments.

We thank the reviewer for their helpful clarification questions and suggestions to improve the manuscript.